# Self-Supervised Dataset Distillation for Transfer Learning

**Dong Bok Lee**[1]*, **Seanie Lee**[1]*, **Joonho Ko**[1], **Kenji Kawaguchi**[2], **Juho Lee**[1], **Sung Ju Hwang**[1]
KAIST[1], National University of Singapore[2]
{markhi, lsnfamily02, joonho.ko, juholee, sjhwang82}@kaist.ac.kr
kenji@comp.nus.edu.sg

## ABSTRACT

Dataset distillation aims to optimize a small set so that a model trained on the set achieves performance similar to that of a model trained on the full dataset. While many supervised methods have achieved remarkable success in distilling a large dataset into a small set of representative samples, however, they are not designed to produce a distilled dataset that can be effectively used to facilitate self-supervised pre-training. To this end, we propose a novel problem of distilling an unlabeled dataset into a set of small synthetic samples for efficient self-supervised learning (SSL). We first prove that a gradient of synthetic samples with respect to a SSL objective in naive bilevel optimization is *biased* due to the randomness originating from data augmentations or masking for inner optimization. To address this issue, we propose to minimize the mean squared error (MSE) between a model's representations of the synthetic examples and their corresponding learnable target feature representations for the inner objective, which does not introduce any randomness. Our primary motivation is that the model obtained by the proposed inner optimization can mimic the *self-supervised target model*. To achieve this, we also introduce the MSE between representations of the inner model and the self-supervised target model on the original full dataset for outer optimization. We empirically validate the effectiveness of our method on transfer learning. Our code is available at https://github.com/db-Lee/selfsup_dd.

## 1 INTRODUCTION

As a consequence of collecting large-scale datasets and recent advances in parallel data processing, deep models have achieved remarkable success in various machine learning problems. However, some applications such as hyperparameter optimization (Franceschi et al., 2017), continual learning (Lopez-Paz & Ranzato, 2017), or neural architecture search (Liu et al., 2019) require repetitive training processes. In such scenarios, it is prohibitively costly to use all the examples from the huge dataset, which motivates the need to compress the full dataset into a small representative set of examples. Recently, many dataset distillation (or condensation) methods (Wang et al., 2018; Zhao et al., 2021; Zhao & Bilen, 2021; Nguyen et al., 2021a;b; Cazenavette et al., 2022; Zhou et al., 2022; Loo et al., 2022; Zhao & Bilen, 2023) have successfully learned a small number of examples on which we can train a model to achieve performance comparable to the one trained on the full dataset.

Despite the recent success of dataset distillation methods, they are not designed to produce a distilled dataset that can be effectively transferred to downstream tasks (Figure 1-(a)). In other words, we may not achieve meaningful performance improvements when pre-training a model on the distilled dataset and fine-tuning it on the target dataset. However, condensing general-purpose datasets into a small set for transfer learning is crucial for some applications. For example, instead of using a large pre-trained model, we may need to search a hardware-specific neural architecture due to constraints on the device (Lee et al., 2021). To evaluate the performance of an architecture during the search process, we repeatedly pre-train a model with the architecture on large unlabeled dataset and fine-tune it on the target training datast, which is time consuming and expensive. If we distill the pre-training dataset into a small dataset at once, we can accelerate the architecture search by pre-training the model on the small set. Another example is target data-free knowledge distillation

---

*Equal contribution

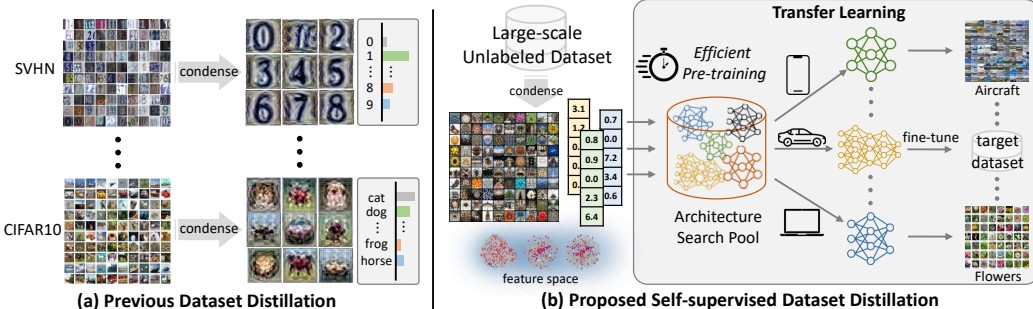

Figure 1: **(a)**: Previous supervised dataset distillation methods. **(b)**: Our proposed method that distills **unlabeled dataset** into a small set that can be effectively used for **pre-training and transfer** to target datasets.

(KD) (Lopes et al., 2017; Raikwar & Mishra, 2022), where we aim to distill a teacher into a smaller student without access to the target training data due to data privacy or intellectual property issues. Instead of the target dataset, we can employ a condensed surrogate dataset for KD (Kim et al., 2023).

To obtain a small representative set for efficient pre-training, as illustrated in Figure 1-(b), we propose a *self-supervised dataset distillation framework* which distills an unlabeled dataset into a small set on which the pre-training will be done. Specifically, we formulate the unsupervised dataset distillation as a bilevel optimization problem: optimizing a small representative set such that a model trained on the small set can induce a latent representation space similar to the space of the model trained on the full dataset. Naively, we can replace the objective function of existing bilevel optimization methods for supervised dataset condensation with a SSL objective function that involves some randomness, such as data augmentations or masking inputs. However, we have empirically found that back-propagating through data augmentation or masking is unstable. Moreover, we prove that a gradient of the self-supervised learning (SSL) loss with randomly sampled data augmentations or masking is *a biased estimator of the true gradient*, explaining the instability.

Based on this insight, we propose to use a mean squared error (MSE) for both inner and outer objective functions, which does not introduce any randomness due to the SSL objectives and thus contributes to stable optimization. First, we parameterize a pair of synthetic examples and target representations of the synthetic ones. For inner optimization, we train a model to minimize the MSE between the target representations and a model's representations of the synthetic examples. Then, we evaluate the MSE between the original data representation of the model trained with the inner optimization and that of the model pre-trained on the original full dataset with a SSL objective. Since we do not sample any data augmentations or input masks, we can avoid the biased gradient of the SSL loss. Lastly, similar to Zhou et al. (2022), we simplify the inner optimization to reduce computational cost. We decompose the model into a feature extractor and a linear head, and optimize only the linear head with kernel ridge regression during the inner optimization while freezing the feature extractor. With the linear head and the frozen feature extractor, we compute the meta-gradient of the synthetic examples and target representations with respect to the outer loss, and update them. To this end, we dub our proposed self-supervised dataset distillation method for transfer learning as ***K**ernel **R**idge **R**egression on **S**elf-supervised **T**arget* (**KRR-ST**).

We empirically show that our proposed KRR-ST significantly outperforms the supervised dataset distillation methods in transfer learning experiments, where we condense a source dataset, which is either CIFAR100 (Krizhevsky et al., 2009), TinyImageNet (Le & Yang, 2015), or ImageNet (Deng et al., 2009), into a small set, pre-train models with different architectures on the condensed dataset, and fine-tune all the models on target labeled datasets such as CIFAR10, Aircraft (Maji et al., 2013), Stanford Cars (Krause et al., 2013), CUB2011 (Wah et al., 2011), Stanford Dogs (Khosla et al., 2011), and Flowers (Nilsback & Zisserman, 2008). Our contributions are as follows:

- We propose a new problem of *self-supervised dataset distillation* for transfer learning, where we distill an unlabeled dataset into a small set, pre-train a model on it, and fine-tune it on target tasks.

- We have observed training instability when utilizing existing SSL objectives in bilevel optimization for self-supervised dataset distillation. Furthermore, we prove that a gradient of the SSL objectives with data augmentations or masking inputs is *a biased estimator of the true gradient*.

- To address the instability, we propose **KRR-ST** using MSE without any randomness at an inner loop. For the inner loop, we minimize MSE between a model representation of synthetic samples and target representations. For an outer loop, we minimize MSE between the original data representation of the model from inner loop and that of the model pre-trained on the original dataset.

- We extensively validate our proposed method on numerous target datasets and architectures, and show that ours outperforms supervised dataset distillation methods.

## 2 RELATED WORK

**Dataset Distillation (or Condensation)** To compress a large dataset into a small set, instead of selecting coresets (Borsos et al., 2020; Mirzasoleiman et al., 2020), dataset condensation optimizes a small number of synthetic samples while preserving information of the original dataset to effectively train high-performing deep learning models. Wang et al. (2018) propose the first dataset distillation (DD) method based on bilevel optimization, where the inner optimization is simply approximated by one gradient-step. Instead of one-step approximation, recent works propose a back-propagation through time for full inner optimization steps (Deng & Russakovsky, 2022), or implicit gradient based on implicit function theorem (Loo et al., 2023). This bilevel formulation, however, incurs expensive computational costs and hinders scaling to large datasets. To overcome this, several papers propose surrogate objective alternatives to bilevel optimization. Specifically, DSA (Zhao et al., 2021; Zhao & Bilen, 2021) minimizes the distance between gradients of original and synthetic samples for each training step. MTT (Cazenavette et al., 2022) proposes to match the parameter obtained on real data and the parameter optimized on the synthetic data. DM (Zhao & Bilen, 2023) matches the first moments of the feature distributions induced by the original dataset and synthetic dataset. As another line of work, Kernel Inducing Points (Nguyen et al., 2021a;b) propose DD methods based on kernel ridge regression, which simplifies inner optimization by back-propagating through Neural Tangent Kernel (NTK) (Lee et al., 2019). Due to the expensive cost of computing NTK for neural networks, FRePo (Zhou et al., 2022) proposes kernel ridge regression on neural features sampled from a pool, and RFAD (Loo et al., 2022) proposes random feature approximation of the neural network Gaussian process. Despite the recent advances in DD methods, none of them have tackled unsupervised/self-supervised DD for transferable learning.

**Self-Supervised Learning** A vast amount of works have proposed self-supervised learning (SSL) methods. A core idea of SSL is that we use large-scale unlabeled data to train a model to learn meaningful representation space that can be effectively transferred to downstream tasks. We introduce a few representative works. SimCLR (Chen et al., 2020a) is one of the representative contrastive learning methods. It maximizes the similarity between two different augmentations of the same input while minimizing the similarity between two randomly chosen pairs. MOCO (He et al., 2020) constructs a dynamic dictionary using a moving average encoder and queue, and minimizes contrastive loss with the dictionary. On the other hand, several non-contrastive works achieve remarkable performance. BYOL (Grill et al., 2020) encodes two different views of an input with a student and teacher encoder, respectively, and minimizes the distance between those two representations. Barlow Twins (Zbontar et al., 2021) constructs a correlation matrix between two different views of a batch of samples with an encoder and trains the encoder to enforce the correlation matrix to the identity matrix. Lastly, MAE (He et al., 2022) learns a meaningful image representation space by masking an image and reconstructing the masked input. In this paper, we utilize Barlow Twins as an SSL framework to train our target model.

## 3 METHOD

### 3.1 PRELIMINARIES

**Problem Definition** Suppose that we are given an unlabeled dataset $X_t = [\mathbf{x}_1 \cdots \mathbf{x}_n]^\top \in \mathbb{R}^{n \times d_x}$ where each row $\mathbf{x}_i \in \mathbb{R}^{d_x}$ is an i.i.d sample. We define the problem of *self-supervised dataset distillation* as the process of creating a compact synthetic dataset $X_s = [\hat{\mathbf{x}}_1 \cdots \hat{\mathbf{x}}_m]^\top \in \mathbb{R}^{m \times d_x}$ that preserves most of the information from the unlabeled dataset $X_t$ for pre-training any neural networks, while keeping $m \ll n$. Thus, after the dataset distillation process, we can transfer knowledge embedded in the large dataset to various tasks using the distilled dataset. Specifically, our final goal is to accelerate the pre-training of a neural network with any architectures by utilizing the distilled dataset $X_s$ in place of the full unlabeled dataset $X_t$ for pre-training. Subsequently, one can evaluate the performance of the neural network by fine-tuning it on various downstream tasks.

**Bilevel Optimization with SSL** Recent success of transfer learning with self-supervised learning (SSL) (Chen et al., 2020a; He et al., 2020; Grill et al., 2020; Zbontar et al., 2021) is deeply rooted in the ability to learn meaningful and task-agnostic latent representation space. Inspired by the SSL,

we want to find a distilled dataset $X_s$ such that a model $\hat{g}_\theta : \mathbb{R}^{d_x} \to \mathbb{R}^{d_y}$, trained on $X_s$ with a SSL objective, achieves low SSL loss on the full dataset $X_t$. Here, $\theta$ denotes the parameter of the neural network $\hat{g}_\theta$. Similar to previous supervised dataset condensation methods (Wang et al., 2018; Zhao et al., 2021; Cazenavette et al., 2022; Deng & Russakovsky, 2022), estimation of $X_s$ can be formulated as a bilevel optimization problem:

$$\underset{X_s}{\text{minimize}} \; \mathcal{L}_{\text{SSL}}(\theta^*(X_s); X_t), \; \text{where} \; \theta^*(X_s) = \arg\min_\theta \mathcal{L}_{\text{SSL}}(\theta; X_s). \tag{1}$$

Here, $\mathcal{L}_{\text{SSL}}(\theta; X_s)$ denotes a SSL loss function with $\hat{g}_\theta$ evaluated on the dataset $X_s$. The bilevel optimization can be solved by iterative gradient-based algorithms. However, it is computationally expensive since computing gradient with respect to $X_s$ requires back-propagating through unrolled computational graphs of inner optimization steps. Furthermore, we empirically find out that back-propagating through data augmentations involved in SSL is unstable and challenging.

## 3.2 KERNEL RIDGE REGRESSION ON SELF-SUPERVISED TARGET

**Motivation** We theoretically analyze the instability of the bilevel formulation for optimizing a condensed dataset with an SSL objective and motivate our objective function. Define $d_\theta$ by $\theta \in \mathbb{R}^{d_\theta}$. Let us write $\mathcal{L}_{\text{SSL}}(\theta; X_s) = \mathbb{E}_{\xi \sim \mathcal{D}}[\ell_\xi(\theta, X_s)]$ where $\xi \sim \mathcal{D}$ is the random variable corresponding to the data augmentation (or input mask), and $\ell_\xi$ is SSL loss with the sampled data augmentation (or input mask) $\xi$. Define $\hat{\theta}(X_s) = \arg\min_\theta \hat{\mathcal{L}}_{\text{SSL}}(\theta; X_s)$ where $\hat{\mathcal{L}}_{\text{SSL}}(\theta; X_s) = \frac{1}{r} \sum_{i=1}^r \ell_{\zeta_i}(\theta, X_s)$ and $\zeta_i \sim \mathcal{D}$. In practice, we compute $\frac{\partial \mathcal{L}_{\text{SSL}}(\hat{\theta}(X_s); X_t)}{\partial X_s}$ to update $X_s$. The use of the empirical estimate $\hat{\mathcal{L}}_{\text{SSL}}(\theta; X_s)$ in the place of the true SSL loss $\mathcal{L}_{\text{SSL}}(\theta; X_s)$ is justified in standard SSL without bilevel optimization because its gradient is always an unbiased estimator of the true gradient: i.e., $\mathbb{E}_\zeta[\frac{\partial \hat{\mathcal{L}}_{\text{SSL}}(\theta; X_s)}{\partial \theta}] = \frac{\partial \mathcal{L}_{\text{SSL}}(\theta; X_s)}{\partial \theta}$ where $\zeta = (\zeta_i)_{i=1}^r$. However, the following theorem shows that this is not the case for bilevel optimization. This explains the empirically observed instability of the SSL loss in bilevel optimization. A proof is deferred to Appendix A.

**Theorem 1.** *The derivative* $\frac{\partial \mathcal{L}_{\text{SSL}}(\hat{\theta}(X_s); X_t)}{\partial X_s}$ *is a biased estimator of* $\frac{\partial \mathcal{L}_{\text{SSL}}(\theta^*(X_s); X_t)}{\partial X_s}$, *i.e.,* $\mathbb{E}_\zeta[\frac{\partial \mathcal{L}_{\text{SSL}}(\hat{\theta}(X_s); X_t)}{\partial X_s}] \neq \frac{\partial \mathcal{L}_{\text{SSL}}(\theta^*(X_s); X_t)}{\partial X_s}$, *unless* $(\frac{\partial \mathcal{L}_{\text{SSL}}(\theta; X_t)}{\partial \theta}|_{\theta=\theta^*(X_s)}) \frac{\partial \theta^*(X_s)}{\partial (X_s)_{ij}} = \mathbb{E}_\zeta[\frac{\partial \mathcal{L}_{\text{SSL}}(\theta; X_t)}{\partial \theta}|_{\theta=\hat{\theta}(X_s)}]\mathbb{E}_\zeta[\frac{\partial \hat{\theta}(X_s)}{\partial (X_s)_{ij}}] + \sum_{k=1}^{d_\theta} \text{Cov}_\zeta[\frac{\partial \mathcal{L}_{\text{SSL}}(\theta; X_t)}{\partial \theta_k}|_{\theta=\hat{\theta}(X_s)}, \frac{\partial \hat{\theta}(X_s)_k}{\partial (X_s)_{ij}}]$ *for all* $(i,j) \in \{1, \ldots, m\} \times \{1, \ldots, d_x\}$.

**Regression on Self-supervised Target** Based on the insight of Theorem 1, we propose to replace the inner objective function with a mean squared error (MSE) by parameterizing and optimizing both synthetic examples $X_s$ and their target representations $Y_s = [\hat{\mathbf{y}}_1 \cdots \hat{\mathbf{y}}_m]^\top \in \mathbb{R}^{m \times d_y}$ as:

$$\mathcal{L}_{\text{inner}}(\theta; X_s, Y_s) = \frac{1}{2} \|Y_s - \hat{g}_\theta(X_s)\|_F^2, \tag{2}$$

which avoids the biased gradient of SSL loss due to the absence of random variables $\zeta$ corresponding to data augmentation (or input mask) in the MSE. Here, $\|\cdot\|_F$ denotes a Frobenius norm. Similarly, we replace the outer objective with the MSE between the original data representation of the model trained with $\mathcal{L}_{\text{inner}}(\theta; X_s, Y_s)$ and that of the target model $g_\phi : \mathbb{R}^{d_x} \to \mathbb{R}^{d_y}$ trained on the full dataset $X_t$ with the SSL objective as follows:

$$\underset{X_s, Y_s}{\text{minimize}} \; \frac{1}{2} \|g_\phi(X_t) - \hat{g}_{\theta^*(X_s, Y_s)}(X_t)\|_F^2, \; \text{where} \; \theta^*(X_s, Y_s) = \arg\min_\theta \mathcal{L}_{\text{inner}}(\theta; X_s, Y_s). \tag{3}$$

Note that we first pre-train the target model $g_\phi$ on the full dataset $X_t$ with the SSL objective, *i.e.,* $\phi = \arg\min_\phi \mathcal{L}_{\text{SSL}}(\phi; X_t)$. After that, $g_\phi(X_t)$ is a fixed target which is considered to be constant during the optimization of $X_s$ and $Y_s$. Here, $g_\phi(X_t) = [g_\phi(\mathbf{x}_1) \cdots g_\phi(\mathbf{x}_n)]^\top \in \mathbb{R}^{n \times d_y}$ and $\hat{g}_{\theta^*(X_s, Y_s)}(X_t) = [\hat{g}_{\theta^*(X_s, Y_s)}(\mathbf{x}_1) \cdots \hat{g}_{\theta^*(X_s, Y_s)}(\mathbf{x}_n)]^\top \in \mathbb{R}^{n \times d_y}$. The intuition behind the objective function is as follows. Assuming that a model trained with an SSL objective on a large-scale dataset generalizes to various downstream tasks (Chen et al., 2020b), we aim to ensure that the representation space of the model $\hat{g}_{\theta^*(X_s, Y_s)}$, trained on the condensed data, is similar to that of the self-supervised target model $g_\phi$.

Again, one notable advantage of using the MSE is that it removes the need for data augmentations or masking inputs for the evaluation of the inner objective. Furthermore, we can easily evaluate

---

**Algorithm 1** Kernel Ridge Regression on Self-supervised Target (KRR-ST)

1: **Input:** Dataset $X_t$, batch size $b$, learning rate $\alpha, \eta$, SSL objective $\mathcal{L}_{\text{SSL}}$, and total steps $T$.
2: Optimize $g_\phi$ with SSL loss on $X_t$ using data augmentation: $\phi \leftarrow \arg\min_\phi \mathcal{L}_{\text{SSL}}(\phi; X_t)$.
3: Initialize $X_s = [\hat{\mathbf{x}}_1 \cdots \hat{\mathbf{x}}_m]^\top$ with a random subset of $X_t$.
4: Initialize $Y_s = [\hat{\mathbf{y}}_1 \cdots \hat{\mathbf{y}}_m]^\top$ with $\hat{\mathbf{y}}_i = g_\phi(\hat{\mathbf{x}}_i)$ for $i = 1, \ldots, m$.
5: Initialize a model pool $\mathcal{M} = \{(\omega_1, W_1, t_1) \ldots, (\omega_l, W_l, t_l)\}$ using $X_s$ and $Y_s$.
6: **while** not converged **do**
7:    Uniformly sample a mini batch $\bar{X}_t = [\bar{\mathbf{x}}_1 \cdots \bar{\mathbf{x}}_b]^\top$ from the full dataset $X_t$.
8:    Uniformly sample an index $i$ from $\{1, \ldots, l\}$ and retrieve the model $(\omega_i, W_i, t_i) \in \mathcal{M}$.
9:    Compute the outer objective $\mathcal{L}_{\text{outer}}(X_s, Y_s)$ with $f_{\omega_i}$ in equation 4.
10:   Update $X_s$ and $Y_s$: $X_s \leftarrow X_s - \alpha\nabla_{X_s}\mathcal{L}_{\text{outer}}(X_s, Y_s)$, $Y_s \leftarrow Y_s - \alpha\nabla_{Y_s}\mathcal{L}_{\text{outer}}(X_s, Y_s)$.
11:   **if** $t_i < T$ **then**
12:      Set $t_i \leftarrow t_i + 1$ and evaluate MSE loss $\mathcal{L}_{\text{MSE}} \leftarrow \frac{1}{2}\|Y_s - h_{W_i} \circ f_{\omega_i}(X_s)\|_F^2$.
13:      Update $\omega_i$ and $W_i$ with $\omega_i \leftarrow \omega_i - \eta\nabla_{\omega_i}\mathcal{L}_{\text{MSE}}$, $W_i \leftarrow W_i - \eta\nabla_{W_i}\mathcal{L}_{\text{MSE}}$.
14:   **else**
15:      Set $t_i \leftarrow 0$ and randomly initialize $\omega_i$ and $W_i$.
16:   **end if**
17: **end while**
18: **Output:** Distilled dataset $(X_s, Y_s)$

---

the inner objective with full batch $X_s$ since the size of $X_s$ (*i.e.*, $m$) is small enough and we do not need $m \times m$ pairwise correlation matrix required for many SSL objectives (Chen et al., 2020a; He et al., 2020; Zbontar et al., 2021; Bardes et al., 2022). Consequently, the elimination of randomness enables us to get an unbiased estimate of the true gradient and contributes to stable optimization.

**Kernel Ridge Regression**  Lastly, following Zhou et al. (2022), we simplify the inner optimization to reduce the computational cost of bilevel optimization in equation 3. First, we decompose the function $\hat{g}_\theta$ into a feature extractor $f_\omega : \mathbb{R}^{d_x} \to \mathbb{R}^{d_h}$ and a linear head $h_W : \mathbf{v} \in \mathbb{R}^{d_h} \mapsto \mathbf{v}^\top W \in \mathbb{R}^{d_y}$, where $W \in \mathbb{R}^{d_h \times d_y}$, and $\theta = (\omega, W)$ (*i.e.*, $\hat{g}_\theta = h_W \circ f_\omega$). Naively, we can train the feature extractor and linear head on $X_s$ and $Y_s$ during inner optimization and compute the meta-gradient of $X_s$ and $Y_s$ w.r.t the outer objective while considering the feature extractor constant. However, previous works (Cazenavette et al., 2022; Zhou et al., 2022; Zhao et al., 2023) have shown that using diverse models at inner optimization is robust to overfitting compared to using a single model.

Based on this insight, we maintain a model pool $\mathcal{M}$ consisting of $l$ different feature extractors and linear heads. To initialize each $h_W \circ f_\omega$ in the pool, we first sample $t \in \{1, \ldots, T\}$ and then optimize $\omega$ and $W$ to minimize the MSE in equation 2 on randomly initialized $X_s$ and $Y_s$ with full-batch gradient descent algorithms for $t$ steps, where $T$ is the maximum number of steps. Afterward, we sample a feature extractor $f_\omega$ from $\mathcal{M}$ for each meta-update. We then optimize another head $h_{W^*}$ on top of the sampled feature extractor $f_\omega$ which is fixed. Here, kernel ridge regression (Murphy, 2012) enables getting a closed form solution of the linear head as $h_{W^*} : \mathbf{v} \mapsto \mathbf{v}^\top f_\omega(X_s)^\top(K_{X_s,X_s} + \lambda I_m)^{-1}Y_s$, where $\lambda > 0$ is a hyperparameter for $\ell_2$ regularization, $I_m \in \mathbb{R}^{m \times m}$ is an identity matrix, and $K_{X_s,X_s} = f_\omega(X_s)f_\omega(X_s)^\top \in \mathbb{R}^{m \times m}$ with $f_\omega(X_s) = [f_\omega(\hat{\mathbf{x}}_1) \cdots f_\omega(\hat{\mathbf{x}}_m)]^\top \in \mathbb{R}^{m \times d}$. Then, we sample a mini-batch $\bar{X}_t = [\bar{\mathbf{x}}_1 \cdots \bar{\mathbf{x}}_b]^\top \in \mathbb{R}^{b \times d_x}$ from the full set $X_t$ and compute a meta-gradient of $X_s$ and $Y_s$ with respect to the following outer objective function:

$$\mathcal{L}_{\text{outer}}(X_s, Y_s) = \frac{1}{2}\left\| g_\phi(\bar{X}_t) - f_\omega(\bar{X}_t)f_\omega(X_s)^\top(K_{X_s,X_s} + \lambda I_m)^{-1}Y_s \right\|_F^2, \quad (4)$$

where $g_\phi(\bar{X}_t) = [g_\phi(\bar{\mathbf{x}}_1) \cdots g_\phi(\bar{\mathbf{x}}_b)]^\top \in \mathbb{R}^{b \times d_y}$ and $f_\omega(\bar{X}_t) = [f_\omega(\bar{\mathbf{x}}_1) \cdots f_\omega(\bar{\mathbf{x}}_b)]^\top \in \mathbb{R}^{b \times d_h}$. Finally, we update the distilled dataset $X_s$ and $Y_s$ with gradient descent algorithms. After updating the distilled dataset, we update the selected feature extractor $f_\omega$ and its corresponding head $h_W$ with the distilled dataset for one step. Based on this, we dub our proposed method as ***Kernel Ridge Regression on Self-supervised Target*** (**KRR-ST**), and outline its algorithmic design in Algorithm 1.

**Transfer Learning**  We now elaborate on how we deploy the distilled dataset for transfer learning scenarios. Given the distilled dataset $(X_s, Y_s)$, we first pre-train a randomly initialized feature extractor $f_\omega$ and head $h_W : \mathbf{v} \in \mathbb{R}^{d_h} \mapsto \mathbf{v}^\top W \in \mathbb{R}^{d_y}$ on the distilled dataset to minimize either MSE for our KRR-ST, KIP (Nguyen et al., 2021a), and FRePO (Zhou et al., 2022), or cross-entropy loss for DSA (Zhao & Bilen, 2021), MTT (Cazenavette et al., 2022), and DM (Zhao & Bilen, 2023):

$$\underset{\omega, W}{\text{minimize}} \frac{1}{2}\|f_\omega(X_s)W - Y_s\|_F^2, \quad \text{or} \quad \underset{\omega, W}{\text{minimize}} \sum_{i=1}^{m} \ell(\hat{\mathbf{y}}_i, \texttt{softmax}(f_\omega(\hat{\mathbf{x}}_i)^\top W)), \quad (5)$$

where $\ell(\mathbf{p}, \mathbf{q}) = -\sum_{i=1}^{c} p_i \log q_i$ for $\mathbf{p} = (p_1, \ldots, p_c), \mathbf{q} = (q_1, \ldots, q_c) \in \Delta^{c-1}$, and $\Delta^{c-1}$ is simplex over $\mathbb{R}^c$. Then, we discard $h_W$ and fine-tune the feature extractor $f_\omega$ with a randomly initialized task-specific head $h_Q : \mathbf{v} \in \mathbb{R}^{d_h} \mapsto \texttt{softmax}(\mathbf{v}^\top Q) \in \Delta^{c-1}$ on a target labeled dataset to minimize the cross-entropy loss $\ell$. Here, $Q \in \mathbb{R}^{d_h \times c}$ and $c$ is the number of classes. Note that we can use any loss function for fine-tuning, however, we only focus on the classification in this work.

## 4 EXPERIMENTS

In this section, we empirically validate the efficacy of our KRR-ST on various applications: transfer learning, architecture generalization, and target data-free knowledge distillation.

### 4.1 EXPERIMENTAL SETUPS

**Datasets**  We use either CIFAR100 (Krizhevsky et al., 2009), TinyImageNet (Le & Yang, 2015), or ImageNet (Deng et al., 2009) as a source dataset for dataset distillation, while evaluating the distilled dataset on CIFAR10 (Krizhevsky et al., 2009), Aircraft (Maji et al., 2013), Stanford Cars (Krause et al., 2013), CUB2011 (Wah et al., 2011), Stanford Dogs (Khosla et al., 2011), and Flowers (Nilsback & Zisserman, 2008). For ImageNet, we resize the images into a resolution of $64 \times 64$, following the previous dataset distillation methods (Zhou et al., 2022; Cazenavette et al., 2022). We resize the images of the target dataset into the resolution of the source dataset, e.g., $32 \times 32$ for CIFAR100 and $64 \times 64$ for TinyImageNet and ImageNet, respectively.

**Baselines**  We compare the proposed method KRR-ST with 8 different baselines including a simple baseline without pre-training, 5 representative supervised baselines from the dataset condensation benchmark (Cui et al., 2022), and 2 kernel ridge regression baselines as follows:

1) **w/o pre**: We train a model on solely the target dataset without any pre-training.

2) **Random**: We pre-train a model on randomly chosen images of the source dataset.

3) **Kmeans** (Cui et al., 2022): Instead of **2) Random** choice, we choose the nearest image for each centroid of kmeans-clustering (Lloyd, 1982) in feature space of the source dataset.

4–6) **DSA** (Zhao & Bilen, 2021), **DM** (Zhao & Bilen, 2023), and **MTT** (Cazenavette et al., 2022): These are representative dataset condensation methods based on surrogate objectives such as gradient matching, distribution matching, and trajectory matching, respectively.

7) **KIP** (Nguyen et al., 2021a;b): Kernel Inducing Points (KIP) is the first proposed kernel ridge regression method for dataset distillation. For transfer learning, we use the distilled datasets with standard normalization instead of ZCA-whitening.

8) **FRePo** (Zhou et al., 2022): Feature Regression with Pooling (FRePo) is a relaxed version of bilevel optimization, where inner optimization is replaced with the analytic solution of kernel ridge regression on neural features. Since FRePo does not provide datasets distilled with standard normalization, we re-run the official code of Zhou et al. (2022) with standard normalization for transfer learning.

**Implementation Details**  Following Nguyen et al. (2021a;b); Zhou et al. (2022), we use convolutional layers consisting of batch normalization (Ioffe & Szegedy, 2015), ReLU activation, and average pooling for distilling a dataset. We choose the number of layers based on the resolution of images, *i.e.*, 3 layers for $32 \times 32$ and 4 layers for $64 \times 64$, respectively. We initialize and maintain $l = 10$ models for the model pool $\mathcal{M}$, and update the models in the pool using full-batch gradient descent with learning rate, momentum, and weight decay being set to $0.1, 0.9$, and $0.001$, respectively. The total number of steps $T$ is set to $1,000$. We meta-update our distilled dataset for $160,000$ iterations using AdamW optimizer (Loshchilov & Hutter, 2019) with an initial learning rate of $0.001$, $0.00001$, and $0.00001$ for CIFAR100, TinyImageNet, and ImageNet, respetively. The learning rate is linearly decayed. We use ResNet18 (He et al., 2016) as a self-supervised target model $g_\phi$ which is trained on a source dataset with Barlow Twins (Zbontar et al., 2021) objective.

After distillation, we pre-train a model on the distilled dataset for $1,000$ epochs with a mini-batch size of $256$ using stochastic gradient descent (SGD) optimizer, where learning rate, momentum, and weight decay are set to $0.1, 0.9$, and $0.001$, respectively. For the baselines, we follow their original

Table 1: The results of **transfer learning with CIFAR100**. The data compression ratio for source dataset is 2%. ConvNet3 is pre-trained on a condensed dataset, and then fine-tuned on target datasets. We report the average and standard deviation over three runs. The best results are bolded.

| | Source | Target | | | | | |
|---|---|---|---|---|---|---|---|
| Method | CIFAR100 | CIFAR10 | Aircraft | Cars | CUB2011 | Dogs | Flowers |
| w/o pre | $64.95_{\pm0.21}$ | $87.34_{\pm0.13}$ | $34.66_{\pm0.39}$ | $19.43_{\pm0.14}$ | $18.46_{\pm0.11}$ | $22.31_{\pm0.22}$ | $58.75_{\pm0.41}$ |
| Random | $65.23_{\pm0.12}$ | $87.55_{\pm0.19}$ | $33.99_{\pm0.45}$ | $19.77_{\pm0.21}$ | $18.18_{\pm0.21}$ | $21.69_{\pm0.18}$ | $59.31_{\pm0.27}$ |
| Kmeans | $65.67_{\pm0.25}$ | $87.67_{\pm0.09}$ | $35.08_{\pm0.69}$ | $20.02_{\pm0.44}$ | $18.12_{\pm0.15}$ | $21.86_{\pm0.18}$ | $59.58_{\pm0.18}$ |
| DSA | $65.48_{\pm0.21}$ | $87.21_{\pm0.12}$ | $34.38_{\pm0.17}$ | $19.59_{\pm0.25}$ | $18.08_{\pm0.33}$ | $21.90_{\pm0.24}$ | $58.50_{\pm0.04}$ |
| DM | $65.47_{\pm0.12}$ | $87.64_{\pm0.20}$ | $35.24_{\pm0.64}$ | $20.13_{\pm0.33}$ | $18.68_{\pm0.33}$ | $21.87_{\pm0.23}$ | $59.89_{\pm0.57}$ |
| MTT | $65.92_{\pm0.18}$ | $87.87_{\pm0.08}$ | $36.11_{\pm0.27}$ | $21.42_{\pm0.03}$ | $18.94_{\pm0.41}$ | $22.82_{\pm0.02}$ | $60.88_{\pm0.45}$ |
| KIP | $65.97_{\pm0.02}$ | $87.90_{\pm0.19}$ | $37.67_{\pm0.36}$ | $23.12_{\pm0.69}$ | $20.10_{\pm0.57}$ | $23.83_{\pm0.13}$ | $63.04_{\pm0.33}$ |
| FRePo | $65.64_{\pm0.40}$ | $87.67_{\pm0.22}$ | $35.34_{\pm0.85}$ | $21.05_{\pm0.06}$ | $18.88_{\pm0.20}$ | $22.90_{\pm0.29}$ | $60.35_{\pm0.07}$ |
| KRR-ST | $\mathbf{66.81}_{\pm0.11}$ | $\mathbf{88.72}_{\pm0.11}$ | $\mathbf{41.54}_{\pm0.37}$ | $\mathbf{28.68}_{\pm0.32}$ | $\mathbf{25.30}_{\pm0.37}$ | $\mathbf{26.39}_{\pm0.08}$ | $\mathbf{67.88}_{\pm0.18}$ |

Table 2: The results of **transfer learning with TinyImageNet**. The data compression ratio for source dataset is 2%. ConvNet4 is pre-trained on a condensed dataset, and then fine-tuned on target datasets. We report the average and standard deviation over three runs. The best results are bolded.

| | Source | Target | | | | | |
|---|---|---|---|---|---|---|---|
| Method | TinyImageNet | CIFAR10 | Aircraft | Cars | CUB2011 | Dogs | Flowers |
| w/o pre | $49.57_{\pm0.18}$ | $88.74_{\pm0.10}$ | $43.81_{\pm0.56}$ | $23.42_{\pm0.16}$ | $22.19_{\pm0.27}$ | $24.74_{\pm0.49}$ | $59.48_{\pm0.37}$ |
| Random | $50.49_{\pm0.30}$ | $88.46_{\pm0.18}$ | $43.17_{\pm0.34}$ | $24.74_{\pm0.36}$ | $21.91_{\pm0.08}$ | $25.17_{\pm0.28}$ | $62.49_{\pm0.40}$ |
| Kmeans | $50.69_{\pm0.37}$ | $88.58_{\pm0.16}$ | $43.77_{\pm0.31}$ | $26.26_{\pm0.33}$ | $22.52_{\pm0.37}$ | $25.85_{\pm0.25}$ | $63.63_{\pm0.38}$ |
| DSA | $50.42_{\pm0.05}$ | $88.77_{\pm0.13}$ | $43.63_{\pm0.13}$ | $26.02_{\pm0.84}$ | $22.98_{\pm0.48}$ | $26.52_{\pm0.33}$ | $63.98_{\pm0.42}$ |
| DM | $49.81_{\pm0.10}$ | $88.48_{\pm0.08}$ | $42.14_{\pm0.38}$ | $25.68_{\pm0.44}$ | $22.48_{\pm0.62}$ | $25.05_{\pm0.28}$ | $63.45_{\pm0.14}$ |
| MTT | $50.92_{\pm0.18}$ | $89.20_{\pm0.05}$ | $48.21_{\pm0.30}$ | $30.35_{\pm0.22}$ | $25.95_{\pm0.14}$ | $28.53_{\pm0.26}$ | $66.07_{\pm0.18}$ |
| FRePo | $49.71_{\pm0.06}$ | $88.28_{\pm0.09}$ | $47.59_{\pm0.59}$ | $29.25_{\pm0.41}$ | $24.81_{\pm0.66}$ | $27.68_{\pm0.37}$ | $62.91_{\pm0.47}$ |
| KRR-ST | $\mathbf{51.86}_{\pm0.24}$ | $\mathbf{89.31}_{\pm0.08}$ | $\mathbf{58.83}_{\pm0.30}$ | $\mathbf{49.26}_{\pm0.64}$ | $\mathbf{35.55}_{\pm0.66}$ | $\mathbf{35.78}_{\pm0.46}$ | $\mathbf{71.16}_{\pm0.70}$ |

Table 3: The results of **transfer learning with ImageNet**. The data compression ratio for the source dataset is $\approx0.08\%$. ConvNet4 is pre-trained on a condensed dataset and then fine-tuned on target datasets. We report the average and standard deviation over three runs. The best results are bolded.

| Method | CIFAR10 | CIFAR100 | Aircraft | Cars | CUB2011 | Dogs | Flowers |
|---|---|---|---|---|---|---|---|
| w/o pre | $88.66_{\pm0.09}$ | $66.62_{\pm0.32}$ | $42.45_{\pm0.46}$ | $23.62_{\pm0.70}$ | $22.00_{\pm0.09}$ | $24.59_{\pm0.46}$ | $59.39_{\pm0.29}$ |
| Random | $88.46_{\pm0.09}$ | $65.97_{\pm0.08}$ | $40.09_{\pm0.46}$ | $20.92_{\pm0.42}$ | $19.41_{\pm0.28}$ | $23.08_{\pm0.40}$ | $56.81_{\pm0.44}$ |
| FRePo | $87.88_{\pm0.20}$ | $65.23_{\pm0.47}$ | $39.03_{\pm0.35}$ | $20.00_{\pm0.73}$ | $19.26_{\pm0.21}$ | $22.05_{\pm0.45}$ | $52.50_{\pm0.51}$ |
| KRR-ST | $\mathbf{89.33}_{\pm0.19}$ | $\mathbf{68.04}_{\pm0.22}$ | $\mathbf{57.17}_{\pm0.16}$ | $\mathbf{46.95}_{\pm0.37}$ | $\mathbf{35.66}_{\pm0.56}$ | $\mathbf{35.51}_{\pm0.45}$ | $\mathbf{70.45}_{\pm0.34}$ |

experimental setup to pre-train a model on their condensed dataset. For fine-tuning, all the experimental setups are fixed as follows: we use the SGD optimizer with learning rate of 0.01, momentum of 0.9 and weight decay of 0.0005. We fine-tune the models for 10,000 iterations (CIFAR100, CIFAR10, and TinyImagenet), or 5,000 iterations (Aircraft, Cars, CUB2011, Dogs, and Flowers) with a mini-batch size of 256. The learning rate is decayed with cosine scheduling.

## 4.2 EXPERIMENTAL RESULTS AND ANALYSIS

**Transfer learning** We investigate how our proposed KRR-ST can be effectively used for transfer learning. To this end, we pre-train a model on the distilled source dataset and fine-tune the model using a target training dataset. We report the average and standard deviation of the model's accuracy on the target test dataset over three runs. First, we use ConvNet3 (3-layer CNN) to distill the CIFAR100 dataset into 1,000 synthetic examples, which is equivalent to 2% of the original dataset. After distillation, we pre-train the model with the synthetic samples and fine-tune it on the target training datasets. As shown in Table 1, KRR-ST outperforms all the baselines, including those using labels for distillation. Next, we distill TinyImageNet into 2,000 synthetic images, which constitute 2% of the original dataset. We pre-train ConvNet4 on the distilled dataset and fine-tune the model on the target datasets. As shown in Table 2, we observe that our unsupervised dataset distillation method outperforms all the baselines by a larger margin than in the previous experiments with

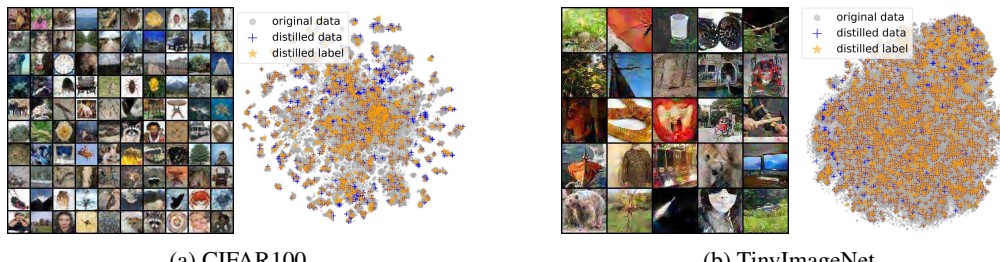

(a) CIFAR100                                    (b) TinyImageNet

Figure 2: **Visualization** of the distilled images, their feature representation and corresponding distilled labels in the output space of the target model. All distilled images are provided in Appendix B.

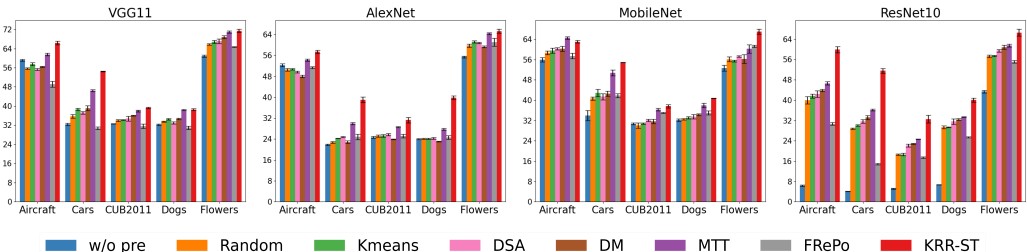

Figure 3: The results of **architecture generalization**. ConvNet4 is utilized for condensing TinyImageNet into 2,000 synthetic examples. Models with different architectures are pre-trained on the condensed dataset and fine-tuned on target datasets. We report the average and standard deviation over three runs. The above results are reported as a tabular format in Appendix C.

the distilled CIFAR100 dataset. Lastly, we distill ImageNet into $1,000$ synthetic samples which are approximately $0.08\%$ of the original full dataset using ConvNet4, and report experimental results in Table 3. For ImageNet experiments, FRePo is the only available supervised dataset distillation method since we can not run the other baselines due to their large memory consumption. These experimental results demonstrate the efficacy of our method in condensing an unlabeled dataset for pre-training a model that can be transferred to target datasets.

**Visualization**   In this paragraph, we analyze how our method distills images and their corresponding target representations in both pixel and feature space. We first present the distilled images $X_s$ and visualize their representation $g_\phi(X_s) \in \mathbb{R}^{m \times 512}$ along with the learned target representation $Y_s \in \mathbb{R}^{m \times 512}$ and representations of the original full data $g_\phi(X_t) \in \mathbb{R}^{n \times 512}$, where $g_\phi$ is the target ResNet18 trained with SSL Barlow Twins objective on the original dataset $X_t$. For visualization, we employ t-SNE (Van der Maaten & Hinton, 2008) to project the high-dimensional representations to 2D vectors. Figure 2 demonstrates the distilled images and corresponding feature space of CIFAR100 and TinyImageNet. As reported in Zhou et al. (2022), we have found that distilled images with our algorithm result in visually realistic samples, which is well-known to be a crucial factor for architecture generalization. Lastly, we observe that the distilled data points cover most of the feature space induced by the full dataset, even with either $1,000$ or $2,000$ synthetic samples which are only $2\%$ of the full dataset. All distilled images are provided in Appendix B.

**Architecture Generalization**   To examine whether our method can produce a distilled dataset that can be generalized to different architectures, we perform the following experiments. First, we use ConvNet4 as $\hat{g}_\theta$ in equation 3 to condense TinyImageNet into 2,000 synthetic examples. Then, we pre-train models of VGG11 (Simonyan & Zisserman, 2015), AlexNet (Krizhevsky et al., 2012), MobileNet (Howard et al., 2017), and ResNet10 (Gong et al., 2022) architectures on the condensed dataset. Finally, the models are fine-tuned on five target datasets — Stanford Cars, Stanford Dogs, Aircraft, CUB2011, and Flowers dataset. We choose those architectures since they are lightweight and suitable for small devices, and pre-trained weights of those architectures for $64 \times 64$ resolution are rarely available on the internet. As shown in Figure 3 and Tables 5 to 8 from Appendix C, our method achieves significant improvements over baselines across different architectures except for one setting (MobileNet on the Aircraft dataset). These results showcase that our method can effectively distill the source dataset into a small one that allows pre-training models with different architectures.

Table 4: The results of **target data-free KD** on CIFAR10. ConvNet4 is utilized for condensing TinyImageNet into 2,000 synthetic examples. Models with different architectures are pre-trained on the condensed dataset and fine-tuned on CIFAR10 using KD loss. We report the average and standard deviation over three runs.

| Method | ConvNet4 | VGG11 | AlexNet | MobileNet | ResNet10 |
|---|---|---|---|---|---|
| Gaussian | $32.45_{\pm0.65}$ | $33.25_{\pm1.33}$ | $30.58_{\pm0.56}$ | $23.96_{\pm0.94}$ | $24.83_{\pm1.86}$ |
| Random | $49.98_{\pm0.73}$ | $51.47_{\pm1.10}$ | $51.16_{\pm0.22}$ | $44.27_{\pm0.92}$ | $40.63_{\pm0.54}$ |
| Kmeans | $52.00_{\pm1.10}$ | $53.86_{\pm0.92}$ | $53.31_{\pm0.65}$ | $48.10_{\pm0.46}$ | $40.90_{\pm0.42}$ |
| DSA | $45.64_{\pm1.28}$ | $47.97_{\pm0.13}$ | $47.42_{\pm0.70}$ | $39.67_{\pm0.92}$ | $37.33_{\pm1.70}$ |
| DM | $46.90_{\pm0.18}$ | $48.41_{\pm0.34}$ | $48.61_{\pm0.25}$ | $40.09_{\pm1.30}$ | $39.35_{\pm0.40}$ |
| MTT | $49.62_{\pm0.90}$ | $53.18_{\pm0.72}$ | $51.22_{\pm0.38}$ | $44.48_{\pm1.04}$ | $38.75_{\pm0.47}$ |
| FRePo | $45.25_{\pm0.40}$ | $50.51_{\pm0.50}$ | $47.24_{\pm0.21}$ | $42.98_{\pm0.71}$ | $42.16_{\pm0.81}$ |
| KRR-ST | $\mathbf{58.31}_{\pm0.94}$ | $\mathbf{62.15}_{\pm1.08}$ | $\mathbf{59.80}_{\pm0.69}$ | $\mathbf{54.13}_{\pm0.49}$ | $\mathbf{52.68}_{\pm1.47}$ |

**Target Data-Free Knowledge Distillation** One of the most challenging transfer learning scenarios is data-free knowledge distillation (KD) (Lopes et al., 2017; Yin et al., 2020; Raikwar & Mishra, 2022), where we aim to distill the knowledge of teacher into smaller student models without a target dataset due to data privacy or intellectual property issues. Inspired by the success of KD with a surrogate dataset (Orekondy et al., 2019; Kim et al., 2023), we utilize distilled TinyImageNet dataset $X_s$ as a surrogate dataset for KD instead of using the target dataset CIFAR10. Here, we investigate the efficacy of each dataset distillation method on this target data-free KD task. First, we are given a teacher model $T_\psi : \mathbb{R}^{d_x} \to \Delta^{c-1}$ which is trained on the target dataset CIFAR10, where $c = 10$ is the number of classes. We first pre-train a feature extractor $f_\omega$, as demonstrated in equation 5. After that, we randomly initialize the task head $h_Q : \mathbf{v} \in \mathbb{R}^{d_h} \mapsto \mathrm{softmax}(\mathbf{v}^\top Q) \in \Delta^{c-1}$ with $Q \in \mathbb{R}^{d_h \times c}$, and fine-tune $\omega$ and $Q$ with the cross-entropy loss $\ell$ using the teacher $T_\psi$ as a target:

$$\underset{\omega, Q}{\mathrm{minimize}} \frac{1}{m} \sum_{i=1}^{m} \ell\left(T_\psi(\hat{\mathbf{x}}_i), \mathrm{softmax}(f_\omega(\hat{\mathbf{x}}_i)^\top Q)\right). \tag{6}$$

In preliminary experiments, we have found that direct use of distilled dataset $X_s$ for KD is not beneficial due to the discrepancy between the source and target dataset. To address this issue, we always use a mean and standard deviation of current mini-batch for batch normalization in both student and teacher models, even at test time, as suggested in Raikwar & Mishra (2022). We optimize the parameter $\omega$ and $Q$ of the student model for $1,000$ epochs with a mini-batch size of $512$, using an AdamW optimizer with a learning rate of $0.0001$. Besides the supervised dataset distillation baselines, we introduce another baseline (Raikwar & Mishra, 2022) referred to as "Gaussian", which uses Gaussian noise as an input to the teacher and the student for computing the KD loss in equation 6, *i.e.*, $\hat{\mathbf{x}}_i \sim \mathcal{N}(\mathbf{0}, I_{d_x})$. Table 4 presents the results of target data-free KD experiments on CIFAR10. Firstly, we observe that utilizing a condensed surrogate dataset is more effective for knowledge distillation than using a Gaussian noise. Moreover, supervised dataset distillation methods (DSA, DM, MTT, and FRePO) even perform worse than the baseline Random. On the other hand, our proposed KRR-ST consistently outperforms all the baselines across different architectures, which showcases the effectiveness of our method for target data-free KD.

## 5 CONCLUSION

In this work, we proposed a novel problem of unsupervised dataset distillation where we distill an unlabeled dataset into a small set of synthetic samples on which we pre-train a model on, and fine-tune the model on the target datasets. Based on a theoretical analysis that the gradient of the synthetic samples with respect to existing SSL loss in naive bilevel optimization is biased, we proposed minimizing the mean squared error (MSE) between a model's representation of the synthetic samples and learnable target representations for the inner objective. Based on the motivation that the model obtained by the inner optimization is expected to imitate the self-supervised target model, we also introduced the MSE between representations of the inner model and those of the self-supervised target model on the original full dataset for outer optimization. Finally, we simplify the inner optimization by optimizing only a linear head with kernel ridge regression, enabling us to reduce the computational cost. The experimental results demonstrated the efficacy of our self-supervised data distillation method in various applications such as transfer learning, architecture generalization, and target data-free knowledge distillation.

**Reproducibility Statement**    We use Pytorch (Paszke et al., 2019) to implement our self-supervised dataset distillation method, KRR-ST. First, we have provided the complete proof of Theorem 1 in Appendix A. Moreover, we have detailed our method in Algorithm 1 and specified all the implementation details including hyperparameters in Section 4.1.  Our code is available at https://github.com/db-Lee/selfsup_dd.

**Ethics Statement**    Our work is less likely to bring about any negative societal impacts. However, we should be careful about bias in the original dataset, as this bias may be transferred to the distilled dataset. On the positive side, we can significantly reduce the search cost of NAS, which, in turn, can reduce the energy consumption when running GPUs.

**Acknowledgement**    This work was supported by Institute of Information & communications Technology Planning & Evaluation (IITP) grant funded by the Korea government(MSIT) (No.2019-0-00075, Artificial Intelligence Graduate School Program(KAIST)), Institute of Information & communications Technology Planning & Evaluation (IITP) grant funded by the Korea government(MSIT) (No.2020-0-00153, and No.2022-0-00713), KAIST-NAVER Hypercreative AI Center, Samsung Electronics (IO201214-08145-01), the National Research Foundation Singapore under the AI Singapore Programme (AISG Award No: AISG2-TC-2023-010-SGIL) and the Singapore Ministry of Education Academic Research Fund Tier 1 (Award No: T1 251RES2207).

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

## A    PROOF OF THEOREM 1

*Proof.* Let $(i,j) \in \{1,\ldots,m\} \times \{1,\ldots,d_x\}$. By the chain rule,

$$\frac{\partial \mathcal{L}_{\text{SSL}}(\theta^*(X_s); X_t)}{\partial (X_s)_{ij}} = \left( \frac{\partial \mathcal{L}_{\text{SSL}}(\theta; X_t)}{\partial \theta} \Big|_{\theta = \theta^*(X_s)} \right) \frac{\partial \theta^*(X_s)}{\partial (X_s)_{ij}} \tag{7}$$

Similarly,

$$\frac{\partial \mathcal{L}_{\text{SSL}}(\hat{\theta}(X_s); X_t)}{\partial (X_s)_{ij}} = \left( \frac{\partial \mathcal{L}_{\text{SSL}}(\theta; X_t)}{\partial \theta} \Big|_{\theta = \hat{\theta}(X_s)} \right) \frac{\partial \hat{\theta}(X_s)}{\partial (X_s)_{ij}}$$

By taking expectation and using the definition of the covariance,

$$\mathbb{E}_\zeta \left[ \frac{\partial \mathcal{L}_{\text{SSL}}(\hat{\theta}(X_s); X_t)}{\partial (X_s)_{ij}} \right] = \mathbb{E}_\zeta \left[ \left( \frac{\partial \mathcal{L}_{\text{SSL}}(\theta; X_t)}{\partial \theta} \Big|_{\theta = \hat{\theta}(X_s)} \right) \frac{\partial \hat{\theta}(X_s)}{\partial (X_s)_{ij}} \right]$$

$$= \mathbb{E}_\zeta \left[ \sum_{k=1}^{d_\theta} \left( \frac{\partial \mathcal{L}_{\text{SSL}}(\theta; X_t)}{\partial \theta_k} \Big|_{\theta = \hat{\theta}(X_s)} \right) \frac{\partial \hat{\theta}(X_s)_k}{\partial (X_s)_{ij}} \right]$$

$$= \sum_{k=1}^{d_\theta} \mathbb{E}_\zeta \left[ \left( \frac{\partial \mathcal{L}_{\text{SSL}}(\theta; X_t)}{\partial \theta_k} \Big|_{\theta = \hat{\theta}(X_s)} \right) \frac{\partial \hat{\theta}(X_s)_k}{\partial (X_s)_{ij}} \right]$$

$$= \sum_{k=1}^{d_\theta} \mathbb{E}_\zeta[v_k]\mathbb{E}_\zeta[\alpha_k] + \sum_{k=1}^{d_\theta} \text{Cov}_\zeta[v_k, \alpha_k],$$

where $v_k = \frac{\partial \mathcal{L}_{\text{SSL}}(\theta; X_t)}{\partial \theta_k}\big|_{\theta = \hat{\theta}(X_s)}$ and $\alpha_k = \frac{\partial \hat{\theta}(X_s)_k}{\partial (X_s)_{ij}}$. By defining the vectors $v = [v_1, v_2, \ldots, v_{d_\theta}]^\top \in \mathbb{R}^{d_\theta}$ and $\alpha = [\alpha_1, \alpha_2, \ldots, \alpha_{d_\theta}]^\top \in \mathbb{R}^{d_\theta}$,

$$\mathbb{E}_\zeta \left[ \frac{\partial \mathcal{L}_{\text{SSL}}(\hat{\theta}(X_s); X_t)}{\partial (X_s)_{ij}} \right] = \sum_{k=1}^{d_\theta} \mathbb{E}_\zeta[v_k]\mathbb{E}_\zeta[\alpha_k] + \sum_{k=1}^{d_\theta} \text{Cov}_\zeta[v_k, \alpha_k]$$

$$= [\mathbb{E}_\zeta[v_1] \quad \cdots \quad \mathbb{E}_\zeta[v_{d_\theta}]] \begin{bmatrix} \mathbb{E}_\zeta[\alpha_1] \\ \vdots \\ \mathbb{E}_\zeta[\alpha_{d_\theta}] \end{bmatrix} + \sum_{k=1}^{d_\theta} \text{Cov}_\zeta[v_k, \alpha_k]$$

$$= \mathbb{E}_\zeta [[v_1 \quad \cdots \quad v_{d_\theta}]] \mathbb{E}_\zeta \left[ \begin{bmatrix} \alpha_1 \\ \vdots \\ \alpha_{d_\theta} \end{bmatrix} \right] + \sum_{k=1}^{d_\theta} \text{Cov}_\zeta[v_k, \alpha_k]$$

$$= \mathbb{E}_\zeta [v]^\top \mathbb{E}_\zeta [\alpha] + \sum_{k=1}^{d_\theta} \text{Cov}_\zeta[v_k, \alpha_k].$$

Therefore, $\mathbb{E}_\zeta \left[ \frac{\partial \mathcal{L}_{\text{SSL}}(\hat{\theta}(X_s); X_t)}{\partial (X_s)_{ij}} \right] = \mathbb{E}_\zeta [v]^\top \mathbb{E}_\zeta [\alpha] + \sum_{k=1}^{d_\theta} \text{Cov}_\zeta[v_k, \alpha_k]$. Comparing this with equation 7 proves the statement. $\qquad\square$

# B VISUALIZATION OF DISTILLED IMAGES

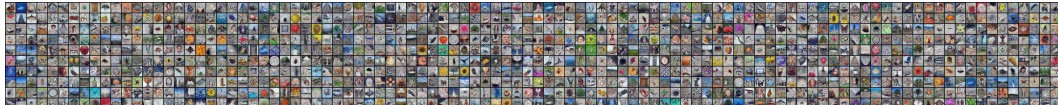

Figure 4: Visualization of the synthetic images distilled by our method in CIFAR100.

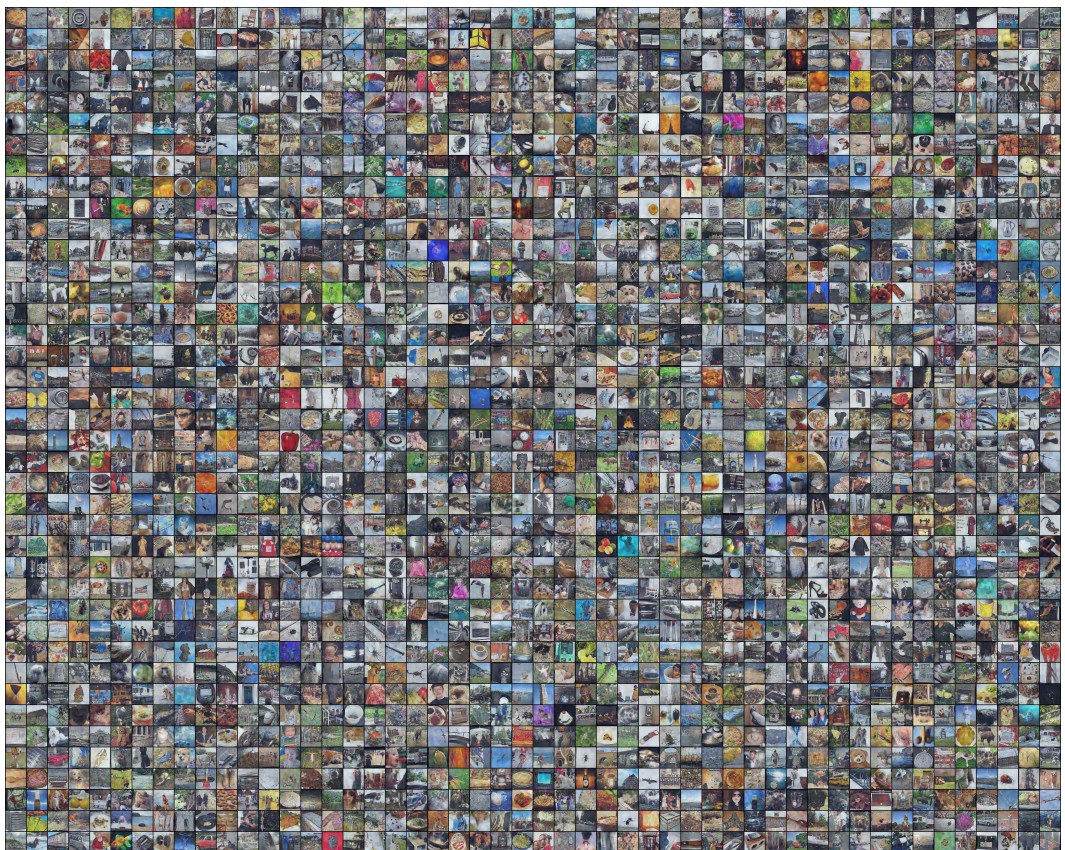

Figure 5: Visualization of the synthetic images distilled by our method in TinyImageNet.

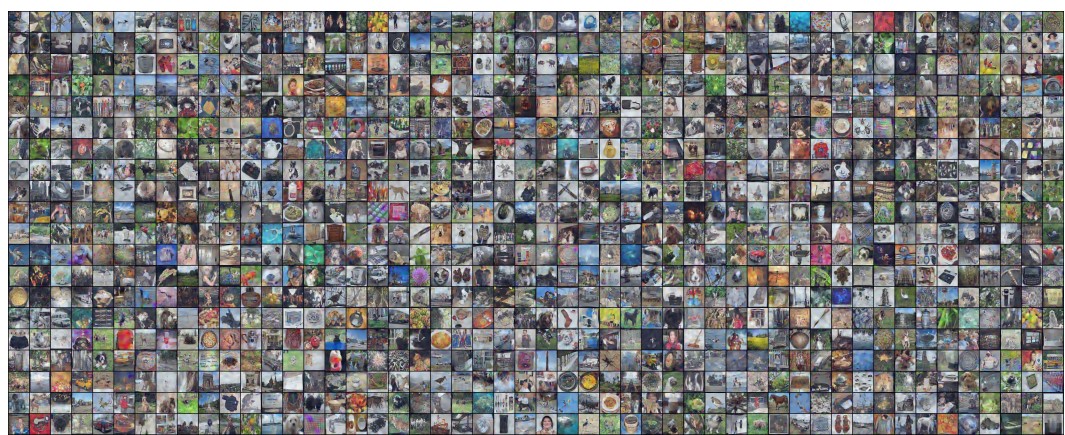

Figure 6: Visualization of the synthetic images distilled by our method in ImageNet.

## C EXPERIMENTAL RESULTS OF ARCHITECTURE GENERALIZATION

Table 5: The results of transfer learning using **VGG11**. Conv4 is utilized for condensing TinyImageNet into 2,000 synthetic examples. We report the average and standard deviation over three runs.

| Method | Aircraft | Cars | CUB2011 | Dogs | Flowers |
|---|---|---|---|---|---|
| w/o pre | $59.17_{\pm 0.39}$ | $32.41_{\pm 0.44}$ | $32.62_{\pm 0.11}$ | $32.24_{\pm 0.34}$ | $60.88_{\pm 0.48}$ |
| Random | $55.67_{\pm 0.27}$ | $35.70_{\pm 0.71}$ | $33.99_{\pm 0.41}$ | $33.55_{\pm 0.21}$ | $65.75_{\pm 0.25}$ |
| Kmeans | $57.50_{\pm 0.65}$ | $38.66_{\pm 0.45}$ | $34.20_{\pm 0.10}$ | $34.50_{\pm 0.33}$ | $66.79_{\pm 0.59}$ |
| DSA | $55.26_{\pm 0.39}$ | $37.22_{\pm 0.54}$ | $34.75_{\pm 0.98}$ | $33.01_{\pm 0.46}$ | $67.07_{\pm 0.99}$ |
| DM | $56.40_{\pm 0.18}$ | $39.13_{\pm 0.99}$ | $36.01_{\pm 0.12}$ | $34.65_{\pm 0.37}$ | $68.82_{\pm 0.60}$ |
| MTT | $61.54_{\pm 0.51}$ | $46.44_{\pm 0.42}$ | $37.95_{\pm 0.33}$ | $38.35_{\pm 0.24}$ | $70.97_{\pm 0.50}$ |
| FRePo | $49.13_{\pm 1.23}$ | $30.72_{\pm 0.51}$ | $31.57_{\pm 0.92}$ | $30.93_{\pm 0.66}$ | $64.66_{\pm 0.18}$ |
| KRR-ST | $\mathbf{66.35}_{\pm 0.76}$ | $\mathbf{54.43}_{\pm 0.17}$ | $\mathbf{39.22}_{\pm 0.28}$ | $\mathbf{38.54}_{\pm 0.31}$ | $\mathbf{71.46}_{\pm 0.70}$ |

Table 6: The results of transfer learning using **AlexNet**. Conv4 is utilized for condensing TinyImageNet into 2,000 synthetic examples. We report the average and standard deviation over three runs.

| Method | Aircraft | Cars | CUB2011 | Dogs | Flowers |
|---|---|---|---|---|---|
| w/o pre | $52.29_{\pm 0.55}$ | $21.83_{\pm 0.29}$ | $24.69_{\pm 0.31}$ | $23.95_{\pm 0.19}$ | $55.46_{\pm 0.33}$ |
| Random | $50.56_{\pm 0.53}$ | $22.77_{\pm 0.34}$ | $25.20_{\pm 0.39}$ | $24.09_{\pm 0.20}$ | $59.81_{\pm 0.62}$ |
| Kmeans | $50.72_{\pm 0.26}$ | $24.23_{\pm 0.05}$ | $25.29_{\pm 0.49}$ | $24.11_{\pm 0.19}$ | $61.20_{\pm 0.45}$ |
| DSA | $49.67_{\pm 0.38}$ | $24.78_{\pm 0.22}$ | $25.77_{\pm 0.38}$ | $24.30_{\pm 0.38}$ | $60.80_{\pm 0.22}$ |
| DM | $48.01_{\pm 0.43}$ | $22.84_{\pm 0.46}$ | $23.88_{\pm 0.14}$ | $23.04_{\pm 0.12}$ | $59.33_{\pm 0.33}$ |
| MTT | $54.24_{\pm 0.45}$ | $29.93_{\pm 0.48}$ | $28.59_{\pm 0.17}$ | $27.74_{\pm 0.34}$ | $64.40_{\pm 0.35}$ |
| FRePo | $51.39_{\pm 0.31}$ | $24.86_{\pm 1.08}$ | $25.19_{\pm 0.72}$ | $24.68_{\pm 0.70}$ | $61.08_{\pm 1.58}$ |
| KRR-ST | $\mathbf{58.76}_{\pm 0.48}$ | $\mathbf{41.60}_{\pm 0.08}$ | $\mathbf{33.69}_{\pm 0.22}$ | $\mathbf{31.55}_{\pm 0.26}$ | $\mathbf{65.50}_{\pm 0.29}$ |

Table 7: The results of transfer learning using **MobileNet**. Conv4 is utilized for condensing TinyImageNet into 2,000 synthetic examples. We report the average and standard deviation over three runs.

| Method | Aircraft | Cars | CUB2011 | Dogs | Flowers |
|---|---|---|---|---|---|
| w/o pre | $55.99_{\pm 0.85}$ | $33.96_{\pm 2.03}$ | $30.73_{\pm 0.39}$ | $32.21_{\pm 0.51}$ | $52.67_{\pm 1.11}$ |
| Random | $58.71_{\pm 0.75}$ | $40.69_{\pm 0.62}$ | $30.03_{\pm 1.10}$ | $32.52_{\pm 0.27}$ | $56.17_{\pm 0.92}$ |
| Kmeans | $59.59_{\pm 1.02}$ | $42.90_{\pm 1.43}$ | $30.82_{\pm 0.24}$ | $33.09_{\pm 0.45}$ | $55.52_{\pm 0.38}$ |
| DSA | $60.32_{\pm 0.47}$ | $41.44_{\pm 1.23}$ | $32.10_{\pm 0.48}$ | $33.58_{\pm 0.88}$ | $57.27_{\pm 0.30}$ |
| DM | $60.29_{\pm 0.97}$ | $42.63_{\pm 1.03}$ | $31.66_{\pm 0.81}$ | $34.39_{\pm 0.37}$ | $56.23_{\pm 1.70}$ |
| MTT | $\mathbf{64.54}_{\pm 0.55}$ | $50.84_{\pm 1.13}$ | $36.31_{\pm 0.51}$ | $37.97_{\pm 0.86}$ | $60.25_{\pm 1.68}$ |
| FRePo | $57.48_{\pm 1.05}$ | $41.85_{\pm 0.76}$ | $35.03_{\pm 0.22}$ | $35.05_{\pm 0.83}$ | $61.29_{\pm 0.40}$ |
| KRR-ST | $63.08_{\pm 0.49}$ | $\mathbf{54.89}_{\pm 0.06}$ | $\mathbf{37.65}_{\pm 0.63}$ | $\mathbf{40.81}_{\pm 0.14}$ | $\mathbf{67.03}_{\pm 1.03}$ |

Table 8: The results of transfer learning using **ResNet10**. Conv4 is utilized for condensing TinyImageNet into 2,000 synthetic examples. We report the average and standard deviation over three runs.

| Method | Aircraft | Cars | CUB2011 | Dogs | Flowers |
|---|---|---|---|---|---|
| w/o pre | $6.34_{\pm 0.25}$ | $4.16_{\pm 0.11}$ | $5.19_{\pm 0.23}$ | $6.69_{\pm 0.11}$ | $43.38_{\pm 0.49}$ |
| Random | $39.96_{\pm 1.48}$ | $28.80_{\pm 0.33}$ | $18.55_{\pm 0.33}$ | $29.34_{\pm 0.61}$ | $57.29_{\pm 0.49}$ |
| Kmeans | $41.67_{\pm 0.78}$ | $30.12_{\pm 0.28}$ | $18.63_{\pm 0.56}$ | $29.34_{\pm 0.11}$ | $57.46_{\pm 0.18}$ |
| DSA | $42.43_{\pm 1.26}$ | $31.66_{\pm 0.70}$ | $22.17_{\pm 0.62}$ | $31.58_{\pm 1.12}$ | $59.47_{\pm 0.54}$ |
| DM | $43.86_{\pm 0.47}$ | $33.17_{\pm 0.81}$ | $22.83_{\pm 0.18}$ | $32.42_{\pm 0.39}$ | $60.85_{\pm 0.76}$ |
| MTT | $46.62_{\pm 0.63}$ | $36.11_{\pm 0.36}$ | $24.65_{\pm 0.08}$ | $33.42_{\pm 0.14}$ | $61.47_{\pm 0.58}$ |
| FRePo | $30.70_{\pm 0.52}$ | $14.82_{\pm 0.24}$ | $17.41_{\pm 0.29}$ | $25.37_{\pm 0.17}$ | $55.09_{\pm 0.48}$ |
| KRR-ST | $\mathbf{59.91}_{\pm 1.24}$ | $\mathbf{51.55}_{\pm 0.90}$ | $\mathbf{32.55}_{\pm 1.55}$ | $\mathbf{40.05}_{\pm 0.74}$ | $\mathbf{66.61}_{\pm 1.38}$ |

# D IMAGENETTE EXPERIMENTS

Table 9: The results of transfer learning using **Conv5**. Conv5 is utilized for condensing ImageNette (Howard, 2019) into 10 synthetic examples. Here, we use ResNet50 pre-trained on ImageNet using Barlow Twins as the target model $g_\phi$ for KRR-ST. We report the average and standard deviation of test accuracy over three runs.

| | Source | | | Target | | |
|---|---|---|---|---|---|---|
| Method | ImageNette | Aircraft | Cars | CUB2011 | Dogs | Flowers |
| w/o pre | $77.55_{\pm0.89}$ | $46.65_{\pm0.26}$ | $16.31_{\pm0.30}$ | $18.92_{\pm0.43}$ | $19.64_{\pm0.40}$ | $46.80_{\pm0.69}$ |
| Random | $62.22_{\pm3.12}$ | $21.15_{\pm1.54}$ | $5.77_{\pm0.91}$ | $7.03_{\pm1.10}$ | $10.10_{\pm0.52}$ | $22.31_{\pm1.34}$ |
| DSA | $63.11_{\pm0.99}$ | $23.07_{\pm1.19}$ | $6.37_{\pm0.50}$ | $7.89_{\pm0.91}$ | $11.12_{\pm0.37}$ | $24.70_{\pm1.71}$ |
| DM | $67.69_{\pm2.39}$ | $28.50_{\pm2.87}$ | $7.47_{\pm0.45}$ | $8.73_{\pm1.17}$ | $11.56_{\pm0.39}$ | $26.23_{\pm2.43}$ |
| KRR-ST | $\mathbf{82.64}_{\pm0.62}$ | $\mathbf{54.34}_{\pm0.22}$ | $\mathbf{16.65}_{\pm0.03}$ | $\mathbf{22.15}_{\pm0.01}$ | $\mathbf{23.29}_{\pm0.12}$ | $\mathbf{47.86}_{\pm0.05}$ |

Table 10: The results of transfer learning using **ResNet18**. Conv5 is utilized for condensing ImageNette (Howard, 2019) into 10 synthetic examples. Here, we use ResNet50 pre-trained on ImageNet using Barlow Twins as the target model $g_\phi$ for KRR-st. We report the average and standard deviation of test accuracy over three runs.

| | Source | | | Target | | |
|---|---|---|---|---|---|---|
| Method | ImageNette | Aircraft | Cars | CUB2011 | Dogs | Flowers |
| w/o pre | $77.16_{\pm0.12}$ | $6.69_{\pm0.18}$ | $4.18_{\pm0.12}$ | $5.02_{\pm0.30}$ | $5.24_{\pm0.06}$ | $41.51_{\pm0.63}$ |
| Random | $79.16_{\pm0.48}$ | $12.47_{\pm0.21}$ | $5.35_{\pm0.06}$ | $6.44_{\pm0.08}$ | $9.94_{\pm0.42}$ | $45.50_{\pm0.29}$ |
| DSA | $78.94_{\pm0.08}$ | $8.22_{\pm0.30}$ | $5.01_{\pm0.11}$ | $6.47_{\pm0.13}$ | $7.82_{\pm0.50}$ | $45.36_{\pm0.21}$ |
| DM | $79.13_{\pm0.16}$ | $8.79_{\pm0.75}$ | $5.21_{\pm0.17}$ | $6.71_{\pm0.14}$ | $8.96_{\pm0.52}$ | $44.71_{\pm0.14}$ |
| KRR-ST | $\mathbf{86.94}_{\pm0.21}$ | $\mathbf{33.94}_{\pm3.15}$ | $\mathbf{6.19}_{\pm0.17}$ | $\mathbf{14.43}_{\pm0.18}$ | $\mathbf{17.26}_{\pm0.21}$ | $\mathbf{51.88}_{\pm0.23}$ |

# E ABLATION STUDY ON THE INITIALIZATION OF $Y_s$

Table 11: The results of ablation study on the initialization of the target representation $Y_s$ with **Conv3**. **w/o init** denotes the initialization of $Y_s = [\hat{\mathbf{y}}_1 \cdots \hat{\mathbf{y}}_m]^\top$ with $\hat{\mathbf{y}}_i$ drawn from standard normal distribution (i.e., $\hat{\mathbf{y}}_i \overset{\text{iid}}{\sim} N(0, I_{d_y})$) for $i = 1, \ldots, m$, and **w init** denotes initializing it with the target model (i.e., $\hat{\mathbf{y}}_i = g_\phi(\hat{\mathbf{x}}_i)$ for $i = 1, \ldots, m$) as done in Algorithm 1, respectively. We report the average and standard deviation of test accuracy over three runs.

| | Source | Target | | | | | |
|---|---|---|---|---|---|---|---|
| Method | CIFAR100 | CIFAR10 | Aircraft | Cars | CUB2011 | Dogs | Flowers |
| **w/o init** | $65.09_{\pm0.09}$ | $87.56_{\pm0.19}$ | $25.54_{\pm0.36}$ | $17.75_{\pm0.07}$ | $16.68_{\pm0.18}$ | $20.82_{\pm0.30}$ | $54.94_{\pm0.15}$ |
| **w init** | $66.46_{\pm0.12}$ | $88.87_{\pm0.05}$ | $32.94_{\pm0.21}$ | $23.92_{\pm0.09}$ | $24.38_{\pm0.18}$ | $23.39_{\pm0.24}$ | $64.28_{\pm0.16}$ |

# F FID SCORE OF CONDENSED DATASET

Table 12: The FID score (Heusel et al., 2017) of condensed dataset.

| Method | CIFAR100 | TinyImageNet |
|---|---|---|
| Random | 34.49 | 20.31 |
| DSA | 113.62 | 90.88 |
| DM | 206.09 | 130.06 |
| MTT | 105.11 | 114.33 |
| KRR-ST | 76.28 | 47.54 |

