# OpenReview forum: "Self-Supervised Dataset Distillation for Transfer Learning"
_ICLR.cc/2024/Conference — ICLR 2024 poster_

### Official Review · Reviewer_fRLb · 2023-10-29

**Soundness:** 3 good
**Presentation:** 3 good
**Contribution:** 2 fair
**Rating:** 6
**Confidence:** 3

**Summary:**

This paper explored a new direction of dataset distillation in the scenario of self-supervised learning, where the target dataset to be compressed were unlabeled. Specifically, the authors first analyzed the problem of naive bi-level formulation with a SSL objective and then proposed to replace it with a MSE loss to mimic the target model. Experiments were conducted to compare the proposed method KRR-ST with other counterparts via transfer learning and results demonstrated the effectiveness of KRR-ST.

**Strengths:**

- The area of dataset distillation for self-supervised learning is under-explored and this paper made an attempt to compress those datasets without labels and provided important insights on future research.
- Rigorous analysis about the biased gradient estimator of the inner optimization with a SSL objective was presented in the paper, and it motivated the proposed MSE loss in KRR-ST without impact of data augmentation.
- Comprehensive experiments were conducted covering three different datasets (CIFAR100, TinyImageNet, and ImageNet) to showcase the advantages of KRR-ST for distillation in self-supervised learning settings.

**Weaknesses:**

- In Equation (2), both $X_s$ and $Y_s$ were trainable parameters and $\theta$ was minimized in the inner optimization. The process looked a little weird to me although I understand that this design was to make inner and outer optimization consistent. More detailed analysis of $Y_s$ can be provided. For example, we know that $X_s$ was initialized with real images, then how about $Y_s$? Would initialization have great impact on the final performance?
- An important comparison with the model pre-trained on the whole dataset was missing. These results should be included to indicate how far current dataset distillation methods were from the "oracle" model and to evaluate the practicality of KRR-ST in real world applications.
- From the visualization, it seemed that the distilled images were not very different from original ones and looked exactly the same. Is it possible to report other metrics to measure the difference between synthetic examples and real ones apart from visualization?
- It was not clear how dataset distillation methods such as DSA and DM were adapted to the self-supervised settings and it was a non-trivial process.

**Questions:**

See questions and suggestions in the Weaknesses above.

---

> ### Author Response · Authors · 2023-11-15
> **Author Response to Reviewer fRLb [1/3]**
>
> Thank you for your time and constructive comments. We response your concerns and questions as follows:
>
> ---
>
> **[Q1]** More detailed analysis of $Y\_s$ can be provided.
>
> > The target representations $Y\_s=[\hat{\mathbf{y}}\_1 \cdots \hat{\mathbf{y}}\_m]^\top$ are  learnable parameters that are optimized along with the distilled images $X\_s=[\hat{\mathbf{x}}\_1 \cdots \hat{\mathbf{x}}\_m]^\top\$.  We aim to optimize the pair of $X_s$ and $Y_s$ such that a representation space of the model $\hat{g}\_\theta$, which is  trained to minimize $\frac{1}{2}\lVert \hat{g}\_\theta(X_s)- Y_s\rVert^2\_F = \frac{1}{2}\sum\_{i=1}^m\lVert \hat{g}\_\theta(\hat{\mathbf{x}}\_i) - \hat\{\mathbf{y}}\_i\rVert^2\_2$, is similar to that of the model  trained on the full dataset with SSL objective.
>
> > In other words, each target representation $\hat{\mathbf{y}}\_i$ serves a **response variable** of the corresponding $\hat{\mathbf{x}}\_i$ such that we want to minimize $\lVert \hat{g}\_\theta(\hat{\mathbf{x}}\_i) - \hat{\mathbf{y}}\_i\rVert\_2^2$ for inner optimization. Then we evaluate the mode $\hat{g}\_\theta$ resulting from the inner optimization on the outer objective, compute the gradient with respect to $X_s$ and $Y_s$, and update $X_s$ and $Y_s$.
>
> > After dataset distillation, we can expect that a model trained with the optimized  $X_s$ and $Y_s$ will obtain a representation space similar to that  of the model trained with the full dataset $X_t$.
>
> ---
>
> **[Q2]** How do we initialize $Y_s=[\hat{\mathbf{y}}_1 \cdots \hat{\mathbf{y}}_m]^\top$?
>
> > Thank you for asking a clarification. We randomly choose a subset of the full dataset $X_t$ for initializing the distilled images $X_s=[\hat{\mathbf{x}}\_1\cdots \hat{\mathbf{x}}\_m]^\top$ and initialize each target representation $\hat{\mathbf{y}}\_i$ with the target model representation $g_\phi(\hat{\mathbf{x}}\_i)$ for $i=1,\ldots, m$. We have clarified how we initialize both $X_s$ and $Y_s$ in Algorithm 1.
>
> ---
>
> **[Q3]** Does the initialization of $Y_s$ matter for the final performance?
>
> > Thank you for your important question on our work. Following your suggestion, we conduct an ablation study by initializing the target representation $Y_s=[\hat{\mathbf{y}}\_1 \cdots \hat{\mathbf{y}}\_m]^\top$ with $\hat{\mathbf{y}}\_i$ drawn from standard normal distribution (i.e., $\hat{\mathbf{y}}\_i \overset{\mathrm{iid}}{\sim} N(0, I\_{d_y})$) for $i=1,\ldots, m$ instead of initializing it with the target model (i.e., $\hat{\mathbf{y}}_i = g\_\phi(\hat{\mathbf{x}}\_i)$ for $i=1,\ldots,m$). We use CIFAR100 as source dataset, and the results are as follows:
> | Method            | CIFAR100       | CIFAR10        | Aircraft       | Cars           | CUB2011        | Dogs           | Flowers        |
> |-------------------|----------------|----------------|----------------|----------------|----------------|----------------|----------------|
> | KRR-ST (**w/o init**)   | $65.09\pm0.09$ | $87.56\pm0.19$ | $25.54\pm0.36$ | $17.75\pm0.07$ | $16.68\pm0.18$ | $20.82\pm0.30$ | $54.94\pm0.15$  |
> | KRR-ST (**w init**) | $66.46\pm0.12$ | $88.87\pm0.05$ | $32.94\pm0.21$ | $23.92\pm0.09$ | $24.38\pm0.18$ | $23.39\pm0.24$ | $64.28\pm0.16$ |
>
> > We observe that the initialization with the target model representation is crucial for the final performance. We have included these results in Appendix E. This mirrors the observation where the neural network pre-trained with self-supervised learning methods tends to converge to better solution than the model with random initialization.
>
> > Note that many **supervised dataset distillation methods [7, 9, 10] also employ specific techniques for initializing distilled class labels** that needs to be optimized along with distilled images. For example, FRePo [7] initializes the distilled class labels with mean-centered one-hot vectors scaled by $\frac{1}{\sqrt{C/10}}$, where $C$ is the number of classes (please see the implementation of FRePo in this **[link](https://github.com/yongchao97/FRePo/blob/43e028a5839a5de367701b9e9544a08ffccb3166/lib/datadistillation/frepo.py#L76)**).
>
> ---

---

> > ### Comment · Reviewer_fRLb · 2023-11-22
> >
> > I appreciate the authors' efforts in preparing this response. I would keep my original score.

---

> > > ### Author Response · Authors · 2023-11-23
> > > **Author Reply**
> > >
> > > Thank you for your acknowledgement. If you have any remaining concerns, please let us know. We are more than happy to address them.
> > >
> > > Best regards,
> > > Authors

---

> ### Author Response · Authors · 2023-11-15
> **Author Response to Reviewer fRLb [2/3]**
>
> ---
>
> **[Q4]** The performance of the oracle model should be included to indicate how far current dataset distillation methods including KRR-ST are from the oracle.
>
> > Thank you for the suggestion. Training a model (oracle) with self-supervised learning on the full dataset is prohibitively time-consuming. Please understand that we may not be able to complete all the experiments during this rebuttal. We instead provide the following surrogate oracle model, $\hat{g}_\theta$, obtained as follows:
> $$\begin{align*}\text{minimize}\_\theta \frac{1}{2} \sum\_{i=1}^n\lVert \hat{g}\_\theta(\mathbf{x}\_i) - g\_\phi (\mathbf{x}\_i)\rVert^2_2\end{align*},$$  where we obtain the surrogate oracle (student) model $\hat{g}\_\theta$ by distilling the target (teacher) model $g\_\phi$ on the entire unlabeled dataset $X_t=[\mathbf{x}_1 \cdots \mathbf{x}_n]^\top$. We then fine-tune the surrogate oracle model $\hat{g}\_\theta$ on the downstream datasets and report the performance as follows:
> | Source       | CIFAR10        | CIFAR100       | Aircraft       | Cars           | CUB2011        | Dogs           | Flowers        |   |   |
> |--------------|----------------|----------------|----------------|----------------|----------------|----------------|----------------|---|---|
> | CIFAR100     | $89.86\pm0.06$ | $69.36\pm0.18$  | $39.38\pm0.95$ | $30.88\pm0.36$ | $27.32\pm0.11$ | $29.16\pm0.10$   | $70.52\pm0.06$ |   |   |
> | TinyImageNet | $89.14\pm0.25$ | $70.13\pm0.11$ | $48.73\pm0.89$ | $48.56\pm0.13$    | $34.99\pm0.20$ | $37.13\pm0.15$ | $69.75\pm0.51$ |   |   |
> | ImageNet     | $91.68\pm0.21$ | $71.58\pm0.14$ | $55.93\pm0.45$ | $50.84\pm0.31$ | $35.80\pm0.07$ | $39.42\pm0.09$ | $77.83\pm0.09$ |   |   |
>
> > We will train the original oracle model and report the performance as soon as possible.
>
> ---
>
> **[Q5]** Is it possible to report other metrics to measure the difference between synthetic examples and real ones?
>
> > Yes, it is. We report the FID [1] between a synthetic dataset and a real dataset below:
> | Method | CIFAR100 | TinyImageNet |
> |--------|----------|--------------|
> | Random | 34.49    | 20.31        |
> | DSA    | 113.62   | 90.88        |
> | DM     | 206.09   | 130.06       |
> | MTT    | 105.11   | 114.33       |
> | KRR-ST | 76.28    | 47.54        |
>
> > Here, we exclude the FID score of FRePo [7] which uses zca normalization since the Inception network [2] used for computing FID score can not process zca normalized images.  Other than Random which is a subset of the original dataset, our distilled dataset is the most similar to the original dataset among the baselines. However, it is hard to say that FID is highly correlated with performance of the distilled dataset. Although Random achieves the best FID, it shows the lowest performance among dataset distillation methods.
>
> > Despite our distilled images appearing natural, their higher FID than Random indicates their distinctiveness from the original dataset. In most realistic scenarios, either choosing  a representative subset of the full dataset may not be optimal or the presence of such a subset is not guaranteed [8]. Thus, a distilled dataset that is directly optimized to compress the full dataset diverges from the randomly chosen subset. We have included these results in Appendix F.
>
> ---
>
> **[Q6]** It is not clear how supervised dataset distillation methods are adapted to the self-supervised settings, which is a non-trivial process.
>
> > Similar to the supervised learning method as a baseline against  self-supervised learning methods in transfer learning scenario [2, 3, 4, 5], we run supervised distillation methods (Kmeans, DSA, DM, MTT, KIP, FRePo)  to distill a source dataset with **its labels, which we do not use for our method**, into a small synthetic dataset. Subsequently, a model is pre-trained on the distilled dataset and fine-tuned on downstream datasets. Further details are provided in the “Implementation Details” section.
>
> ---

---

> ### Author Response · Authors · 2023-11-16
> **Author Response to Reviewer fRLb [3/3]**
>
> ---
>
> **References**
>
> [1] Heusel, Martin, et al. "Gans trained by a two time-scale update rule converge to a local nash equilibrium." Advances in neural information processing systems 30 (2017).
>
> [2] Szegedy, Christian, et al. "Rethinking the inception architecture for computer vision." Proceedings of the IEEE conference on computer vision and pattern recognition. 2016.
>
> [3] Chen, Ting, et al. "A simple framework for contrastive learning of visual representations." International conference on machine learning. PMLR, 2020.
>
> [4] Zbontar, Jure, et al. "Barlow twins: Self-supervised learning via redundancy reduction." International Conference on Machine Learning. PMLR, 2021.
>
> [5] Caron, Mathilde, et al. "Emerging properties in self-supervised vision transformers." Proceedings of the IEEE/CVF international conference on computer vision. 2021.
>
> [6] Ericsson, Linus, Henry Gouk, and Timothy M. Hospedales. "How well do self-supervised models transfer?." Proceedings of the IEEE/CVF Conference on Computer Vision and Pattern Recognition. 2021.
>
> [7] Zhou, Yongchao, Ehsan Nezhadarya, and Jimmy Ba. "Dataset distillation using neural feature regression." Advances in Neural Information Processing Systems 35 (2022): 9813-9827.
>
> [8]  Zhao, Bo, Konda Reddy Mopuri, and Hakan Bilen. "Dataset Condensation with Gradient Matching." International Conference on Learning Representations. 2021.
>
> [9] Nguyen, Timothy, Zhourong Chen, and Jaehoon Lee. "Dataset Meta-Learning from Kernel Ridge-Regression." International Conference on Learning Representations. 2020.
>
> [10] Deng, Zhiwei, and Olga Russakovsky. "Remember the past: Distilling datasets into addressable memories for neural networks." Advances in Neural Information Processing Systems 35 (2022): 34391-34404.

---

### Official Review · Reviewer_hYSv · 2023-10-30

**Soundness:** 3 good
**Presentation:** 3 good
**Contribution:** 3 good
**Rating:** 8
**Confidence:** 3

**Summary:**

The authors introduce a novel problem within the domain of dataset distillation, focusing on the distillation of an unlabeled dataset into a compact set of small synthetic samples tailored for optimizing self-supervised learning (SSL) efficiency.
In this work, the authors address a significant challenge related to the bias introduced during the calculation of gradients for synthetic samples with respect to SSL objectives. This bias originates from the inherent randomness associated with data augmentation and masking techniques. To mitigate this issue, they present an innovative strategy. This approach centers on minimizing the mean squared error (MSE) between a model's representations of synthetic examples and the corresponding learnable target feature representations, effectively removing the randomness from the gradient computation.
In terms of computational efficiency, the authors propose a streamlined methodology that leverages a fixed feature extractor. They focus their optimization efforts on a linear head atop this stable feature extractor, resulting in substantial reductions in computational overhead. Importantly, this linear head optimization is thoughtfully formulated to yield a closed-form solution, employing kernel ridge regression for practical implementation.
The practical utility and impact of the proposed method are rigorously validated through empirical assessments spanning various applications, particularly those involving transfer learning. The experimental results are promising and demonstrate the efficacy of the proposed solution.

**Strengths:**

1. The authors introduce a novel problem within the context of dataset distillation, addressing the challenge of distilling an unlabeled dataset into synthetic samples optimized explicitly for self-supervised learning (SSL). As far as I know, this is the first work in this direction.

2.  The authors identify and effectively tackle a significant bias issue that arises during gradient calculations for synthetic samples. By introducing a methodology that minimizes mean squared error (MSE) between model representations and target feature representations, they successfully eliminate randomness in gradient computations, enhancing the reliability of their approach.

3. A notable strength lies in the authors' approach to computational efficiency. By assuming a fixed feature extractor and optimizing a linear head on top of it, they substantially reduce the computational burden. Furthermore, their formulation allows for a closed-form solution using kernel ridge regression, streamlining the implementation.

4 The writing is clear and well-organized, making it easy for readers to understand the problem, approach, and results. The authors provide a structured narrative that guides the reader through their research.

**Weaknesses:**

1. Clarity Issue: The concept of "target representation" requires further elucidation for clarity and better understanding.

2. Theoretical Analysis of Instability: It is good to conduct a theoretical analysis of the instability inherent in the bilevel formulation when optimizing a condensed dataset with a self-supervised learning (SSL) objective. My question is, does this instability observed in the bilevel formulation similarly apply to the supervised dataset condensation formulation?

3. Rationale for Kernel Ridge Regression: The reason for selecting kernel ridge regression as the methodology in this work warrants clarification. Is there a specific rationale behind this choice within the proposed solution? Could alternative matching strategies such as distribution matching or gradient matching also be integrated into the proposed method?

4. Stability Issues with SSL Data Augmentation: Furthermore, our empirical observations indicate that the process of back-propagating through data augmentations utilized in self-supervised learning (SSL) introduces instability and poses challenges. However, further explanation is needed to fully comprehend why data augmentation in SSL leads to these instabilities.

**Questions:**

Please refer to "Weaknesses" part.

---

> ### Author Response · Authors · 2023-11-15
> **Author Response to Reviewer hYSv [1/2]**
>
> Thank you for your time and constructive comments. We respond to your concerns and questions as follows:
>
> ---
>
> **[Q1]** The concept of “target representation” requires further elucidation for clarity and better understanding.
>
> > The target representations $Y_s=[\hat{\mathbf{y}}\_1 \cdots \hat{\mathbf{y}}\_m]^\top$ are  learnable parameters that are optimized along with the  distilled images $X_s=[\hat{\mathbf{x}}\_1 \cdots \hat{\mathbf{x}}\_m]^\top\$.    We aim to optimize  the pair of $(X_s, Y_s)$  such that a representation space of the model $\hat{g}\_\theta$, which are  trained to minimize $\frac{1}{2}\lVert \hat{g}_{\theta} (X_s)- Y_s\rVert^2_F= \frac{1}{2}\sum\_{i=1}^m \lVert \hat{g}\_\theta (\hat{\mathbf{x}}_i) - \hat\{\mathbf{y}}_i\rVert^2_2 $,  is similar to that of the model, $g\_\phi$,  trained on the full dataset with SSL objective.
>
> > In other words, each target representation $\hat{\mathbf{y}}\_i$ serves a **response variable** of the corresponding $\hat{\mathbf{x}}\_i$ such that we want to minimize  $\lVert \hat{g}_\theta(\mathbf{x}_i) - \hat{\mathbf{y}}_i\rVert_2^2 $ for inner optimization.  Then we evaluate the mode $\hat{g}\_\theta$ resulting from the inner optimization on the outer objective, compute the gradient with respect to $X_s$ and $Y_s$, and update $X_s$ and $Y_s$.
>
> > After dataset distillation, we can expect that a model trained with the optimized  $X_s$ and $Y_s$ will obtain a representation space similar to that  of the model trained with the full dataset $X_t$.
>
> ---
>
> **[Q2]** Does the instability observed in the bilevel formulation with SSL objective apply to the supervised dataset condensation formulation?
>
> > Yes it does. The issue of biased gradient is independent of whether using a label for dataset distillation. If there is any randomness such as data augmentation for inner optimization, the (meta-)gradient of the outer objective is a biased estimator of the true gradient.
>
>  > For supervised objective functions such as cross-entropy loss, we can train a model without data augmentation to eliminate randomness for inner optimization [1]. However this approach is not feasible for self-supervised learning (SSL) objectives. Many of existing SSL objectives [1,2,3] require data augmentation or random masking, introducing the problem of biased gradient for bilevel optimization. Thus, we propose minimizing MSE between representation of distilled images and target representation without any randomness for inner optimization and thus we get an unbiased gradient estimator.
>
> ---
>
> **[Q3]**  Is there a specific rationale behind the choice of kernel ridge regression?
>
> > We chose the kernel ridge regression because of its computational efficiency.  Kernel ridge regression allows us to solve the inner optimization with a closed form, which offers significant speedup and eliminates the need for hessian computation.
>
> > To precisely solve the bilevel optimization problem, however, we need to solve the inner optimization every time we evaluate the outer objective function, and this process is prohibitively slow. Additionally,  it is computationally expensive since it involves computing the hessian for back-propagating through the inner optimization trajectory.
>
> ---
>
>
> **[Q4]** Could alternative matching strategies such as distribution matching or gradient matching also be integrated into the proposed method?
>
> >  It is not straightforward to integrate matching objectives such as distribution matching [6] or gradient matching [5] into our proposed framework since they require class label information. For distribution matching, it computes empirical Maximum Mean Discrepancy (MMD) between class-wise feature representations of real dataset and distilled dataset: $\sum\_{c=1}^C\lVert \frac{1}{n_c} \sum\_{i=1}^{n_c}\hat{g}\_\theta (\hat{\mathbf{x}}^{(c)}\_i) - \frac{1}{m_c} \sum\_{j=1}^{m_c} \hat{g}\_\theta (\mathbf{x}^{(c)}\_j) \rVert_2^2$, where $\mathbf{x}^{(c)}_i$ and $\hat{\mathbf{x}}^{(c)}_j$  denote real image and distilled image of class $c$, respectively.
>
> > Similarly, gradient matching requires to iteratively compute  class-wise distance between gradient of a model trained on original dataset and distilled dataset: $D(\nabla_{\theta} \mathcal{L}(\theta; X^{(c)}\_s), \nabla\_\theta \mathcal{L}(\theta; X^{(c)}\_t) )$ for $c=1, \ldots, C$, where $D$ is a distance metric and $X\_t^{(c)}$ and $X\_s^{(c)}$ denote a set of real images and distilled images of class $c$, respectively.
>
> > Another choice is Reverse Mode Differentiation [7], which computes the meta-gradient of the distilled dataset by backpropagation through time (or inner updates). However, it is notoriously slow due to its computational expensiveness. For example, we found that distilling the CIFAR100 dataset into 1000 images with RMD requires approximately 50 seconds per iteration, while KRR only requires about 0.2 seconds. This is why we chose KRR as our optimization method.
>
> ---

---

> ### Author Response · Authors · 2023-11-15
> **Author Response to Reviewer hYSv [2/2]**
>
> ---
>
> **[Q5]**  Further explanation is needed to fully comprehend why data augmentation is SSL leads to training instability of bilevel optimization.
>
> > **Theorem 1** explains the empirically observed training instability when employing data augmentation for inner optimization. If we sample data augmentation or input mask for inner optimization process, the gradient of the outer objective with respect to distilled images $X_s$ and target representation $Y_s$ becomes a biased estimator. This motivates our proposed inner objective which eliminates any randomness and minimizes MSE between feature representation of distilled images $X_s$ and target representation $Y_s$.
>
> ---
>
> **References**
>
> [1] Deng, Zhiwei, and Olga Russakovsky. "Remember the past: Distilling datasets into addressable memories for neural networks." Advances in Neural Information Processing Systems 35 (2022): 34391-34404.
>
> [2] Chen, Ting, et al. "A simple framework for contrastive learning of visual representations." International conference on machine learning. PMLR, 2020.
>
> [3] Zbontar, Jure, et al. "Barlow twins: Self-supervised learning via redundancy reduction." International Conference on Machine Learning. PMLR, 2021.
>
> [4] He, Kaiming, et al. "Masked autoencoders are scalable vision learners." Proceedings of the IEEE/CVF conference on computer vision and pattern recognition. 2022.
>
> [5] Zhao, Bo, Konda Reddy Mopuri, and Hakan Bilen. "Dataset Condensation with Gradient Matching." Ninth International Conference on Learning Representations 2021. 2021.
>
> [6] Zhao, Bo, and Hakan Bilen. "Dataset condensation with distribution matching." Proceedings of the IEEE/CVF Winter Conference on Applications of Computer Vision. 2023.
>
> [7] Franceschi, Luca, et al. "Forward and reverse gradient-based hyperparameter optimization." International Conference on Machine Learning. PMLR, 2017.

---

### Official Review · Reviewer_4KT8 · 2023-10-31

**Soundness:** 2 fair
**Presentation:** 3 good
**Contribution:** 3 good
**Rating:** 6
**Confidence:** 3

**Summary:**

The paper considers dataset distillation, which aims at learning a small number of representative examples for a large data set. Unlike previous supervised methods, the authors target a self-supervised learning setup. To counter issues with naive bi-level optimization, they adapt the training loss to an MSE-based objective more akin to feature distillation. This allows for further simplifications such as casting one of the subproblems as kernel ridge regression. The method is evaluated in a transfer learning setting where the goal is to use the distilled data set to train a variety of networks which are then transferred to different data sets. A broad range of source data sets, target data sets and network architectures is considered.

**Strengths:**

The paper is overall well written and easy to follow. The method is relatively well-motivated, with the scenario when one wants to train many different architectures to find the best one for mobile/resource constrained deployment. The experiments cover a broad range of source, and transfer data sets, and many different network architectures are considered. The method is novel as far as I can tell (although I’m not an expert on dataset distillation).

**Weaknesses:**

Despite the overall positive impression, I see several weaknesses:
- In the experiments the authors distill the data sets into 1000-2000 examples, for self-supervised learning, without augmentation. The authors do not comment on augmentations when training on the distilled data. This approach might work for the small models and low resolution used in the experiments, but I’m not convinced that it generalizes to larger models, more complex data sets and higher resolution. Data augmentation is a central component in many SSL methods including Barlow Twins, which the authors use.
- Unrelated to data augmentation, I feel it would be necessary to run the algorithm on a less small-scale setup, e.g. on 224x224 ImageNet, and on larger downstream models (ResNet18 or similar) to make a convincing case, in particular given the complexity of the algorithm. I know this requires some compute, but one such experiment would still be necessary in my opinion.
- Some baselines might be weak; for example MobileNet and ResNet10 from scratch get < 4% accuracy on Cars.

Minor comments:
- The abstract might be hard to follow for readers unfamiliar with prior works on dataset distillation.

**Questions:**

- For the kmeans clustering baseline, what does “kmeans-clustering in feature space of the source dataset” mean? What feature space is used?
- Is the Barlow-Twin target model trained with augmentation?

---

> ### Author Response · Authors · 2023-11-15
> **Author Response to Reviewer 4KT8 (1/2)**
>
> Thank you for your time and constructive comments. We respond to your concerns and questions as follows:
>
> ---
>
> **[Q1]** The authors did not use data augmentation, which would not work for larger models, more complex datasets, and higher resolutions.
>
> > First of all, as shown in Theorem 1, the reason why we do not use data augmentation for inner optimization is that the gradient of the outer objective becomes a **biased estimator when employing data augmentation** for the inner optimization. This theoretical analysis aligns with the empirically observed instability associated with the use of data augmentation for inner optimization.
>
> > Secondly, we use the exact same data augmentation as the original paper [1] for training a target model $g_\phi$ on the full dataset with Barlow Twins objective (line 2 in Algorithm 1). We have revised the draft to include the detail in Algorithm 1.
>
>
> > Lastly, in Table 1,2, and 3,  we have already shown that our proposed method **outperforms all the baselines with models (AlexNet, VGG11, MobileNet, ResNet10)  larger than ConvNet, on complex datasets with high resolution images** (Tiny ImageNet and full ImageNet with 64x64 resolution).
>
> ---
>
> **[Q2]** Need large scale experiments such as using 224x224 ImagNet with ResNet18 as a downstream model.
>
> > For a fair comparison, we follow the exact same experimental setup of previous works [3,4], where all images from ImageNet dataset are resized to 64x64 due to computational constraints of the baselines.  Addressing the scalability issue of dataset distillation is left for future work.
>
> > Instead of 224x224 ImageNet, **we have performed dataset distillation of [ImageNette](https://github.com/fastai/imagenette)  with 224x224 resolution, which has not been done by any previous works**. The condensation process involves reducing 9,469 into 10 images,  employing Random, DSA, DM, and KRR-ST methods with Conv5 architecture. The condensed dataset is subsequently evaluated on  both Conv5 and ResNet18 architecture. The results are as follows:
> | Method (Conv5)  | ImageNette     | Aircraft       | Cars           | CUB2011        | Dogs           | Flowers        |   |   |   |
> |---------|----------------|----------------|----------------|----------------|----------------|----------------|---|---|---|
> | w/o pre | $77.55\pm0.89$ | $46.65\pm0.26$ | $16.31\pm0.30$ | $18.92\pm0.43$ | $19.64\pm0.40$ | $46.80\pm0.69$ |   |   |   |
> | Random  | $62.22\pm3.12$ | $21.15\pm1.54$ | $5.77\pm0.91$  | $7.03\pm1.10$  | $10.10\pm0.52$ | $22.31\pm1.34$ |   |   |   |
> | DSA     | $63.11\pm0.99$ | $23.07\pm1.19$ | $6.37\pm0.50$  | $7.89\pm0.91$  | $11.12\pm0.37$ | $24.70\pm1.71$ |   |   |   |
> | DM      | $67.69\pm2.39$ | $28.50\pm2.87$ | $7.47\pm0.45$  | $8.73\pm1.17$  | $11.56\pm0.39$ | $26.23\pm2.43$ |   |   |   |
> | KRR-ST    | $82.64\pm0.62$ | $54.34\pm0.22$ | $16.65\pm0.03$  | $22.15\pm0.01$ | $23.29\pm0.12$ | $47.86\pm0.05$ |   |   |   |
>
> >| Method (ResNet18)  | ImageNette     | Aircraft       | Cars          | CUB011         | Dogs           | Flower         |   |
> |---------|----------------|----------------|---------------|----------------|----------------|----------------|---|
> | w/o pre | $77.16\pm0.12$ | $6.59\pm0.18$  | $4.18\pm0.12$ | $5.02\pm0.30$  | $5.24\pm0.06$  | $41.51\pm0.63$  |   |
> | Random  | $79.16\pm0.48$ | $12.47\pm0.21$ | $5.35\pm0.06$ | $6.44\pm0.08$  | $9.94\pm0.42$  | $45.50\pm0.29$ |   |
> | DSA     | $78.97\pm0.08$ | $8.22\pm0.30$  | $5.01\pm0.11$ | $6.47\pm0.13$  | $7.82\pm0.50$  | $45.36\pm0.21$ |   |
> | DM      | $79.13\pm0.16$ | $8.79\pm0.75$  | $5.21\pm0.17$ | $6.71\pm0.14$  | $8.96\pm0.52$  | $44.71\pm0.14$ |   |
> | KRR-ST  | $86.94\pm0.21$ | $33.93\pm3.15$ | $6.19\pm0.17$ | $14.43\pm0.18$ | $17.26\pm0.21$ | $51.88\pm0.23$ |   |
>
> > We observe that KRR-ST **achieves consistent performance gains** over the models trained from scratch across all the datasets. We have included these results in Appendix D.
>
> ---

---

> ### Author Response · Authors · 2023-11-17
> **Author Response to Reviewer 4KT8 (2/2)**
>
> ---
>
> **[Q3]** The performance of MobileNet and ResNet10 from scratch on Cars are too low.
>
> > We more extensively tune hyperparameters of MobileNet and ResNet10 from scratch on Cars, resulting in notable improvement of test accuracy: MobileNet ($3.94\pm0.31 \rightarrow 7.06\pm0.04$) and ResNet10 ($2.43\pm0.08 \rightarrow 3.50\pm0.05$).
>
> > Note that the Cars dataset is a particularly challenging task for the models without any pre-training due to the small number of training instances (8,144) with 196 classes.
>
> ---
>
> **[Q4]** The abstract might be too hard to follow for readers who are unfamiliar with prior works on dataset distillation.
>
> > We have revised the draft to offer a more detailed background of dataset distillation.
>
> ---
>
> **[Q5]** For the k-means clustering baseline, what does “k-means clustering in feature space of the source dataset” mean? What feature space is used?
>
> > We use the [k-means clustering baseline](https://github.com/justincui03/dc_benchmark)  provided by Cui et a., 2022 [2]. A model is trained on the full dataset with a supervised objective to extract features from each data point. Subsequently, we run k-means clustering on the extracted features with the $\ell_2$ distance metric.
>
> ---
>
> **[Q6]** Is the Barlow-Twins trained with data augmentation?
>
> > Yes, we train the model with data augmentation exactly same as done in the original paper of Barlow-Twins [1]. Based on this, we have revised Algorithm 1.
>
> ---
>
> **References**
>
> [1] Zbontar, Jure, et al. "Barlow twins: Self-supervised learning via redundancy reduction." International Conference on Machine Learning. PMLR, 2021.
>
> [2] Cui, Justin, et al. "DC-BENCH: Dataset condensation benchmark." Advances in Neural Information Processing Systems 35 (2022): 810-822.
>
> [3] Zhou, Yongchao, Ehsan Nezhadarya, and Jimmy Ba. "Dataset distillation using neural feature regression." Advances in Neural Information Processing Systems 35 (2022): 9813-9827.
>
> [4] Cui, Justin, et al. "Scaling up dataset distillation to imagenet-1k with constant memory." International Conference on Machine Learning. PMLR, 2023.

---

### Official Review · Reviewer_EboT · 2023-11-01

**Soundness:** 3 good
**Presentation:** 3 good
**Contribution:** 2 fair
**Rating:** 5
**Confidence:** 4

**Summary:**

This paper targets at a new problem branched from conventional dataset distillation (DD) ------- unsupervised DD, which aims to synthesize an informative small dataset that can be used for facilitating self-supervised pre-training. The experimental results show the potential of the proposed method in the application of transfer learning, architecture generalization and data-free knowledge distillation.

**Strengths:**

1) The problem that this paper focused on is somewhat new to the dataset distillation community;
2) The presentation and writing of this paper is coherent, the idea is easy to follow.

**Weaknesses:**

1) This paper dose NOT choose the state-of-the-art baselines in dataset distillation for comparison, such as IDC, IDM, etc.
2) The authors only provide the experimental results of transfer learning, but did NOT provide the test accuracy of the model trained barely on the distilled dataset. This makes me wondering if the distilled images can keep enough information compared to the images distilled by other baselines, or is the proposed method only performs well in the scenario of transfer learning ?

**Questions:**

See Above.

---

> ### Author Response · Authors · 2023-11-15
> **Author Response to Reviewer EboT**
>
> Thank you for your time and constructive comments. We respond to your concerns and questions as follows:
>
> ---
>
> **[Q1]** The state of the art baselines in dataset distillation such as IDC [1] and IDM [2] are not included.
>
> > Please recall that we have already included a strong baseline, **FRePo [3], which is a more recent model than IDC and outperforms IDM (IDM: 21.9 vs FRePo: 25.4 on TinyImageNet dataset) for supervised dataset distillation task**.
>
> > Furthermore, both IDC [1] and IDM [2] proposed efficient parameterization methods for dataset distillation, which are **orthogonal to our framework**. Specifically, given a storage budget (e.g., $1000\times3\times32\times32$ for CIFAR100), they parameterize low-resolution images ($4000\times3\times16\times16$). These low-resolution images are decoded into higher resolution images  (e.g., $4000\times3\times32\times32$ for CIFAR100) with upsampling functions, **resulting in a larger number of distilled images** compared to other dataset distillation methods. Note that these parameterization methods can also be plugged into our framework for improved performance. We leave this research direction as our future work.
>
> ---
>
> **[Q2]** The authors did not provide the test accuracy of the model trained only on the distilled dataset.
>
> > In our **self-supervised dataset distillation** problem, evaluating the test accuracy of a model trained solely on the distilled dataset is not feasible since we assume that we only have **unlabeled datasets** for dataset distillation. Thus, we can only learn a latent representation $Y_s$ of the distilled images $X_s$.
>
> ---
>
> **[Q3]** Can the distilled images keep enough information compared to the images distilled by other baselines?
>
> > Yes, our distilled images keep enough information of the full dataset compared to other baselines since our method achieves better performance for transfer learning than the others. If distilled images retain meaningful information, then a model pre-trained on the distilled images will more effectively adapt to downstream tasks. This empirical verification is already presented in Table 1, 2, and 3.
>
> ---
>
> **[Q4]** Does the proposed method only perform well in the scenario of transfer learning?
>
> > Our method also performs well in the non-transfer learning scenario, showing the usefulness of our distilled dataset for fine-tuning a model on the same source dataset. After distillation, we pre-train a model on the distilled source dataset and fine-tune the model on the same source dataset as shown in the “Source” column of Table 1 and 2. In these experiments, our method consistently outperforms all the other baselines.
>
> ---
>
> **References**
>
> [1] Kim, Jang-Hyun, et al. "Dataset condensation via efficient synthetic-data parameterization." International Conference on Machine Learning. PMLR, 2022.
>
> [2] Zhao, Ganlong, et al. "Improved distribution matching for dataset condensation." Proceedings of the IEEE/CVF Conference on Computer Vision and Pattern Recognition. 2023.
>
> [3] Zhou, Yongchao, Ehsan Nezhadarya, and Jimmy Ba. "Dataset distillation using neural feature regression." Advances in Neural Information Processing Systems 35 (2022): 9813-9827.

---

### Official Review · Reviewer_59jF · 2023-11-01

**Soundness:** 3 good
**Presentation:** 3 good
**Contribution:** 3 good
**Rating:** 6
**Confidence:** 4

**Summary:**

Dataset distillation is an important and interesting direction in dealing with data-redundancy settings.
Most dataset distillation methods focus on dataset-specific tasks, overlooking the transfer ability of condensed datasets.   This paper tried to address the transfer ability in the unlabeled setting, via self-supervised in a bi-level optimization framework. The distillation experiments verified the proposed method.

**Strengths:**

1. Interesting and novel problem setting:  self-supervised dataset distillation for transfer learning, which might produce task-agnostic condensed data and boost transferability.
2. Theorem contribution: a gradient of the SSL objectives with data augmentations or masking inputs is a biased estimator of the true gradient. And provide detailed proof.
3. Interesting experiments that outperform the supervised distillation method with self-supervised learning.

**Weaknesses:**

1. What is the motivation for minimizing MSE between the original data representation of the model from inner loop and that of the model pre-trained on the original dataset?

2. Why self-supervised learning method is better than the supervised method in this problem? I only see the empirical results, could you provide more explanation?


Update: I have read the responses and my concerns are partially addressed. The authors did not provide more explanation than empirical results (It might be a more difficult problem that is beyond the scope of this paper). However, I still vote to accept this paper regarding its setting and method novelty.

**Questions:**

Please refer to the above weaknesses.

---

> ### Author Response · Authors · 2023-11-15
> **Author Response to Reviewer 59jF**
>
> Thank you for your time and constructive comments. We respond to your concerns and questions as follows:
>
> ---
>
> **[Q1]** What is the motivation for minimizing MSE between the original data representation of the model from the inner loop and that of the pre-trained on the original dataset?
>
> > Please recall that we have already described the motivation behind the outer objective function in Section 3.2:
>
> >"The intuition behind the objective function is as follows. Assuming that a model trained with a SSL objective on a large-scale dataset generalizes to various downstream tasks (Chen et al., 2020b), we aim to ensure that the representation space of the model $\hat{g}_{\theta^{*}(X_s, Y_s)}$ ,  trained on the condensed data, is similar to that of the self-supervised target model $g\_\phi$".
>
> > Moreover, we have empirically validated our motivation by visualizing our distilled dataset in Figure 2. The visualization shows that the distilled data points cover most of the feature space induced by the original dataset.
>
> **[Q2]** Why is a self-supervised learning method better than a supervised method in this problem?
>
> >  Our self-supervised dataset distillation method is based on the recent success of self-supervised learning in transfer learning. Self-supervised learning methods are known to learn features that transfer more effectively to downstream tasks than supervised ones [1, 2, 3]. Our objective function aims to learn a distilled dataset that induces  feature representation space similar to feature space of a model with self-supervised learning on the full dataset. Consequently, our distilled dataset with self-supervised learning outperforms the ones with supervised learning for our applications — transfer learning and target data-free knowledge distillation.
>
> ---
>
> **References**
>
> [1] Caron, Mathilde, et al. "Emerging properties in self-supervised vision transformers." Proceedings of the IEEE/CVF international conference on computer vision. 2021.
>
> [2] Ericsson, Linus, Henry Gouk, and Timothy M. Hospedales. "How well do self-supervised models transfer?." Proceedings of the IEEE/CVF Conference on Computer Vision and Pattern Recognition. 2021.
>
> [3] He, Kaiming, et al. "Masked autoencoders are scalable vision learners." Proceedings of the IEEE/CVF conference on computer vision and pattern recognition. 2022.

---

### Author Response · Authors · 2023-11-17
**General Response**

We express our gratitude to all the reviewers for their constructive comments. We particularly appreciate their recognition of **the novelty in our problem setting (59jF, EboT, hYSv)**, **the innovation in our method (4KT8)** accompanied by a **theoretical analysis (59jF, fRLb)**, **extensive experimental results (4KT8, fRLb)**, and their acknowledgment of **the high quality of our writing (EboT, 4KT8, hYSv)**.
We have responded to the individual comments from the reviewers below and believe that we have successfully responded to most of them. Here, we briefly summarize the revision of our draft requested by reviewers:

- As a response to **Reviewer 4KT8**, we have revised **the abstract** to provide background knowledge of dataset distillation for readers who are not familiar with it.

- As a response to **Reviewer fRLb**, we have included how synthetic images $X_s$ and target representations $Y_s$ are initialized in **Algorithm 1**.

- As a response to **Reviewer 4KT8**, We explicitly specified that the target model is trained with data augmentation in **Algorithm 1**.

- As a response to **Reviewer 4KT8**, We have included experiments on ImageNette with 224 x 224 resolution in **Appendix D**.

- As a response to **Reviewer fRLb**, we have added an ablation study on the initialization of the target representations $Y_s$ in **Appendix E**.

- As a response to**Reviewer fRLb**, we have reported FID between original datasets and distilled datasets in **Appendix F**.

---

### Author Response · Authors · 2023-11-22
**Gentle Reminder**

Thank you for your time and effort in reviewing our work. As the end of the interactive phase approaches (22nd AOE), we would like to take the remaining time to double-check whether we have adequately addressed any concerns mentioned in the initial reviews of our work. If anything is unclear, please feel free to contact us.

Best Regards, Authors.

---

### Author Response · Authors · 2023-11-23
**Gentle reminder - The interactive discussion phase will end in less than 4 hours**

Dear Reviewers,

Thank you all for dedicating your time and effort to reviewing our paper. We would like to remind you that there are only 4 hours left for the interactive discussion. Therefore, we kindly ask you to review our responses and the revisions. We believe our responses and clarifications have addressed any remaining questions, and we are open to addressing any further concerns you may have. Thank you for reviewing our paper once again.

Best,

Authors

---

### Public Comment · ~Siddharth_Joshi1 · 2023-12-12
**Request for distilled sets and other files needed to run code**

Hi authors,

Thank you for this very interesting work!

I was interested in experimenting with your method and was wondering if you could share:
1. the distilled images on CIFAR10, CIFAR100, TinyImageNet etc. that you have distilled using your method.
2. files of the naming convention: X_tr_{size}, Y_tr_{size}, X_te_{size} and Y_te_{size} that are missing in the code submission. While running the code, I saw these files are required by the code and I'm unable to run the distillation without it.

Looking forward to hearing back from you!

Sid.

---

### Meta-Review · Area_Chair_EnVZ · 2023-12-04

**Metareview:**

This paper proposes to distill an unlabelled data set into a smaller synthetic data set specific for self-supervised learning.  The reviewers found the problem relevant and novel, with solid theoretical analysis showing that SSL objectives entail biased gradient estimators for data distillation and the experiments are convincing, although more baselines and architectures could have been considered.

**Justification For Why Not Higher Score:**

Lack of more modern architectures and pushback on reviewer's request to add state-of-the-art baselines.

**Justification For Why Not Lower Score:**

Overall, the paper is interesting and makes an important contribution. The method nicely builds on the theoretical insights and seems to perform well.

---

### Decision · Program_Chairs · 2024-01-16

Accept (poster)